# Can VLMs Reason Robustly?
# A Neuro-Symbolic Investigation

**Weixin Chen**                                               *weixinc2@illinois.edu*
*University of Illinois Urbana-Champaign*

**Antonio Vergari**                                           *avergari@ed.ac.uk*
*University of Edinburgh*

**Han Zhao**                                                  *hanzhao@illinois.edu*
*University of Illinois Urbana-Champaign*

**Reviewed on OpenReview:** *https://openreview.net/forum?id=4y6jiE6Q6O*

## Abstract

Vision-Language Models (VLMs) have been applied to a wide range of reasoning tasks, yet it remains unclear whether they can reason robustly under distribution shifts. In this paper, we study covariate shifts in which the perceptual input distribution changes while the underlying prediction rules do not. To investigate this question, we consider visual deductive reasoning tasks, where a model is required to answer a query given an image and logical rules defined over the object concepts in the image. Empirically, we find that VLMs fine-tuned through gradient-based end-to-end training can achieve high in-distribution accuracy but fail to generalize under such shifts, suggesting that fine-tuning does not reliably induce the underlying reasoning function. This motivates a neuro-symbolic perspective that decouples perception from reasoning. However, we further observe that recent neuro-symbolic approaches that rely on black-box components for reasoning can still exhibit inconsistent robustness across tasks. To address this issue, we propose VLC, a neuro-symbolic method that combines VLM-based concept recognition with circuit-based symbolic reasoning. In particular, task rules are compiled into a symbolic program, specifically a circuit, which executes the rules exactly over the object concepts recognized by the VLM. Experiments on three simple visual deductive reasoning tasks with distinct rule sets show that VLC consistently achieves higher task accuracy on out-of-distribution data than other reasoning paradigms. Code is available at https://github.com/uiuctml/VLC.

## 1 Introduction

Vision-Language Models (VLMs) have recently been applied to a wide range of reasoning tasks, including abstract (Chollet, 2019; Unsal & Akkus, 2025), temporal (Li et al., 2024b; Fu et al., 2025), and document reasoning (Zhu et al., 2024; Wang et al., 2024), as well as relational, attributive, and order understanding (Yüksekgönül et al., 2023a; Zhao et al., 2022). Despite these advances, it remains unclear whether VLMs can reason robustly under distribution shifts. Here, we focus on *covariate shifts* (Shimodaira, 2000; Sugiyama et al., 2007; 2008; Bickel et al., 2009), where the distribution of the perceptual input changes, while the underlying rules for prediction do not. To study the robustness against covariate shifts, we use *visual deductive reasoning tasks*. In these tasks, an image containing multiple objects is provided, along with rules that define the reasoning function based on object concepts. The VLM is prompted to answer a query that requires reasoning over the object concepts using the given rules. See Fig. 1 (left) for an example. We focus on these tasks because they include samples with perceptual inputs of varying complexity and whose

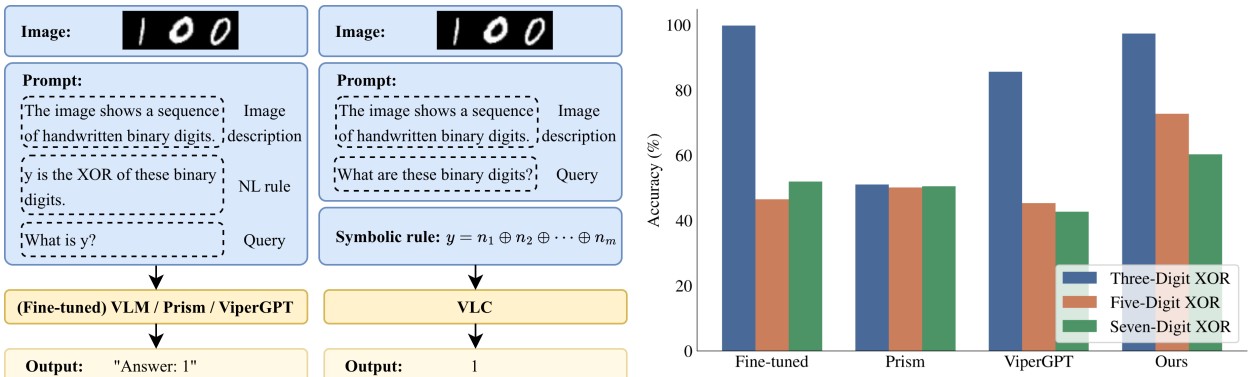

Figure 1: **Left: baseline method pipeline.** Given an image containing multiple objects and a natural-language rule about object concepts, the (fine-tuned) VLM, Prism, and ViperGPT are required to answer a query by reasoning with the given rule based on the image. **Middle: VLC pipeline.** VLC decouples perception from reasoning, compiles the symbolic rule into a circuit, and infers the final answer using the circuit evaluation. **Right: performance of different paradigms on datasets sharing the same reasoning function but differing in the number of objects per image.** The reasoning function is *logical XOR* (see Section 2 for definition), and the three datasets contain images with three, five, and seven handwritten binary digits, respectively.

labels are generated by the same reasoning function. For instance, the reasoning function can be fixed as the logical XOR, while the perceptual input contains different numbers of digits.

Gradient-based end-to-end fine-tuning is the canonical approach for adapting VLMs to specific downstream tasks. Following this standard practice, we fine-tune VLMs on visual deductive reasoning tasks using the causal language modeling loss. As shown in Fig. 1 (right), despite achieving high accuracy on in-distribution data, the fine-tuned models fail to generalize to out-of-distribution (OOD) data that involves a different number of objects but follows the same reasoning function. This result contrasts with the strong generalization often observed when VLMs are fine-tuned for other vision-language tasks like object recognition (Li et al., 2022), suggesting that gradient-based end-to-end fine-tuning does not necessarily enable the model to learn the underlying reasoning function from data. To see whether this limitation stems from the end-to-end reasoning paradigm, we evaluate two recent neuro-symbolic approaches that decouple perception from reasoning. Prism (Qiao et al., 2024) delegates perception to a VLM and reasoning to a Large Language Model (LLM). ViperGPT (Surís et al., 2023) prompts an LLM to generate executable code programs that call upon various specialized pretrained models. However, as we shall see in Section 5.2, even with this decoupling of perception and reasoning, both approaches still show limited robustness under covariate shifts.

This leads to a natural question: *How can we encode a reasoning function into VLMs to achieve robust reasoning?* [1] Inspired by recent studies (Cooper et al., 2025a; Al-Tahan et al., 2024) showing that VLMs excel at recognition tasks but struggle with reasoning, and in line with Qiao et al. (2024); Surís et al. (2023); Kamali & Kordjamshidi (2025), we propose VLC, a neuro-symbolic reasoning paradigm for VLMs that decomposes the end-to-end reasoning process into two sequential phases: *VLM-based concept recognition* and *circuit-based symbolic reasoning*. In the first phase, a VLM serves as the neural module, leveraging its strong recognition capabilities to identify object concepts in the input image. In the second phase, we use circuits (Oztok & Darwiche, 2014; Choi et al., 2020; Vergari et al., 2021) as the symbolic module, which compiles the provided rules into its structure (Darwiche, 2011; Lagniez & Marquis, 2017; Muise et al., 2012). At inference time, the VLM first recognizes object concepts, after which the circuit applies the compiled rules to derive the final answer. By explicitly encoding the true reasoning function into the circuit, VLC ensures interpretable and robust reasoning and allows for principled neuro-symbolic integration where constraints can be embedded in the learning pipeline of deep learning architecture (Manhaeve et al., 2018a; Ahmed et al.,

---

[1]Throughout this paper, we use reasoning robustness to refer specifically to the ability to apply a fixed, explicitly provided deductive rule under controlled covariate shifts in perceptual input, rather than to general-purpose reasoning in open-ended multimodal settings.

2022; Chen et al., 2025a). For our purposes, and striving for simplicity, we adopt the two-stage prediction scenario of Chen et al. (2025a): learning is decomposed into a first stage where the VLM is used to predict object concepts, and then the circuit is used to make predictions, essentially firing the symbolic rules based on the observed concepts.

To evaluate different reasoning paradigms, we advocate going back to benchmarking simple visual deductive reasoning tasks where generalization can be controlled in a rigorous way. Specifically, we use the `rsbench` benchmark suite (Bortolotti et al., 2024) to generate datasets with covariate shifts (Shimodaira, 2000; Sugiyama et al., 2007; 2008; Bickel et al., 2009): the perceptual input varies through the number of objects in the image, while labels are produced by the same reasoning function. In particular, we consider three distinct reasoning functions: *arithmetic addition*, *logical XOR*, and a *relational check*. Note that these datasets differ from classic neuro-symbolic benchmarks, which often assume that each object is already perfectly segmented and provided as a separate image, and typically do not provide concept annotations or explicit rules specifying how object concepts map to the final label. Empirically, even on these simple tasks, VLMs fine-tuned on samples with the fewest objects fail to generalize to samples with more objects, suggesting that gradient-based end-to-end fine-tuning does not reliably learn the underlying reasoning function from data. In contrast, VLC consistently achieves competitive accuracy across all datasets under covariate shifts, indicating that a simple neuro-symbolic pipeline that explicitly encodes the reasoning function as an external symbolic program can effectively improve robustness. In addition, we conduct a series of ablation studies. Notably, we find that scaling up model size improves the performance of VLMs on concept recognition but does not necessarily enhance their reasoning performance, which is consistent with findings in Cherti et al. (2023); Al-Tahan et al. (2024); Zhang et al. (2024a).

Our main contributions are threefold. **(1)** In Section 3, we propose a neuro-symbolic method named VLC for visual deductive reasoning tasks. VLC decomposes end-to-end reasoning into VLM-based concept recognition and circuit-based symbolic reasoning. **(2)** In Section 5, we demonstrate that VLMs do not learn to emulate the underlying reasoning function through gradient-based end-to-end fine-tuning. We further observe that fully-fledged neuro-symbolic approaches, Prism and ViperGPT, which rely on black-box components to complete reasoning, can be unreliable and yield inconsistent robustness across tasks. **(3)** The consistently high robustness of VLC across tasks with distinct reasoning functions, shown in Section 5, highlights the benefit of decoupling perception from reasoning and compiling the reasoning function into a symbolic program.

## 2 Visual Deductive Reasoning

Visual deductive reasoning tasks are designed to test whether a model can deduce the answer to a query from objects present in an image and rules for prediction. A key feature of these tasks is that rules are explicitly provided as part of the input. They differ from classic neuro-symbolic benchmarks (Bortolotti et al., 2024; Manhaeve et al., 2018b; Vermeulen et al., 2023), where each object is often perfectly segmented and provided as a separate image, and the rule specifying the reasoning function is typically not given. Thus, a model has to infer the reasoning function from data and encode it in the learned parameters. Visual deductive reasoning tasks also differ from standard visual question answering (VQA) benchmarks (Yüksekgönül et al., 2023a; Chen et al., 2024a; Hudson & Manning, 2019), where the reasoning function is often sample-specific, highly abstract, and not provided explicitly. For example, a sample may contain a radiograph image and a query like "Which organ appears abnormal in this radiograph?". The distinctive features of visual deductive reasoning tasks allow us to investigate a focused question: *Even when the reasoning rule is explicitly available, can a model capture and apply it reliably under covariate shifts, and what type of approach best supports this behavior?*

Each sample from visual deductive reasoning tasks consists of three input components. First, it contains an image with multiple objects. We construct splits where training and validation images contain fewer objects, while test images contain more objects, creating a controlled covariate shift. Second, the task provides symbolic rules that concisely and precisely specify how object concepts map to the label. As pretrained VLMs may not interpret symbolic rules reliably, equivalent natural-language descriptions are also given. Third, there is a textual query that asks for the value of the label.

In this paper, we consider the following visual deductive reasoning tasks with distinct reasoning functions.

**MNAdd.** Each image contains two rows of handwritten digit images from the MNIST dataset (LeCun et al., 2002), representing two multi-digit numbers. The task requires the model to compute the sum of these two numbers. The reasoning function is therefore the arithmetic addition, and the sum is used as the label $y$. Suppose that each number can be represented using $m$ bits, denoted as $a_{m-1} \ldots a_0$ and $b_{m-1} \ldots b_0$, where $a_{m-1}$ and $b_{m-1}$ are the most significant bits. The reasoning function is formulated as the following symbolic rule:

$$
\begin{aligned}
x_i &= (a_i \wedge \neg b_i) \vee (\neg a_i \wedge b_i), \\
s_i &= (x_i \wedge \neg c_i) \vee (\neg x_i \wedge c_i), \\
c_{i+1} &= (a_i \wedge b_i) \vee (x_i \wedge c_i),
\end{aligned}
\tag{1}
$$

where $i \in \{0, \ldots, m-1\}$ and $c_0 = 0$. Here, $x_i$ denotes the XOR of $a_i$ and $b_i$, $s_i$ denotes the resulting sum bit, and $c_{i+1}$ denotes the carry bit to the next position. The output is an $(m+1)$-bit binary number $c_m s_{m-1} \ldots s_0$, which is then converted to a decimal value $y$. The equivalent natural-language description is: *"Image description: The image shows two rows of handwritten digits. Each row represents a multi-digit number.\nRule: y is the sum of these two numbers.\nQuery: What is y?"*

**MNLogic.** Each image contains a sequence of handwritten binary digit images from the MNIST dataset. The task requires the model to compute the XOR of these digits. Suppose $m$ binary digits are present in an image, denoted as $n_1, \ldots, n_m$. The reasoning function, logical XOR, is formulated as the following symbolic rule:

$$
\begin{aligned}
z_1 &= (n_1 \wedge \neg n_2) \vee (\neg n_1 \wedge n_2), \\
z_i &= (z_{i-1} \wedge \neg n_{i+1}) \vee (\neg z_{i-1} \wedge n_{i+1}), \quad i \in \{2, \ldots, m-1\}, \\
y &= z_{m-1},
\end{aligned}
\tag{2}
$$

which corresponds to $y = n_1 \oplus n_2 \oplus \cdots \oplus n_m$. The equivalent natural-language description would be: *"Image description: The image shows a sequence of handwritten binary digits.\nRule: y is the XOR of these binary digits.\nQuery: What is y?"*

**KandLogic.** Each image contains multiple geometric primitives with various shapes and colors. The task requires the model to determine whether all objects of the same shape have the same color. The reasoning function is therefore a relational check. Suppose there are $m$ geometric primitives in an image. Let the shape and color of the $i$-th primitive be denoted by $s_i$ and $c_i$, respectively, for $i \in 1, \ldots, m$. The reasoning function is formulated as the following symbolic rule:

$$
y = \bigwedge_{1 \leq i < j \leq m} \left( \mathbb{I}(s_i \neq s_j) \vee \mathbb{I}(c_i = c_j) \right),
\tag{3}
$$

where $\mathbb{I}(s_i \neq s_j)$ indicates whether two primitives have different shapes and $\mathbb{I}(c_i = c_j)$ indicates whether they have the same color. This rule is equivalent to $\forall i \neq j, \ \mathbb{I}(s_i = s_j) \rightarrow \mathbb{I}(c_i = c_j)$. The corresponding natural-language description is: *"Image description: The image shows multiple geometric primitives, each with a specific shape and color.\nRule: If all primitives with the same shape have the same color, then y is True. Otherwise, y is False.\nQuery: What is y?"*

## 3 🚧 VLC: Decoupling Perception from Reasoning

Deep neural networks (DNNs) excel at perception. They can extract statistical patterns from raw data, but they often struggle with structured, rule-based reasoning. In contrast, symbolic programs such as circuits excel at reasoning. They can enforce a given logical rule exactly, but they are less flexible when operating on raw, unstructured inputs (Manhaeve et al., 2018a; Ahmed et al., 2022; Chen et al., 2025a). Therefore, we advocate a simple design for visual deductive reasoning that combines the strengths of both. Instead of relying on a single deep model to perform end-to-end reasoning, we decompose the problem into perception and reasoning. Specifically, we introduce a neuro-symbolic reasoning paradigm, VLC, which uses a VLM

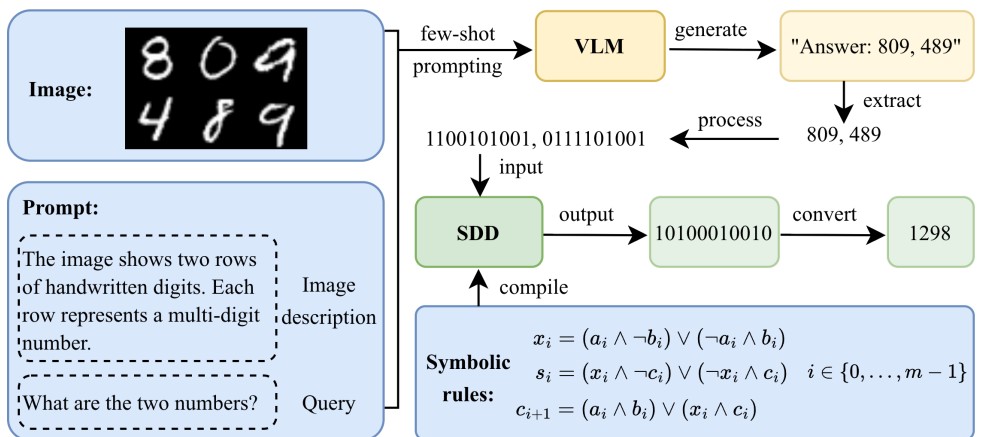

Figure 2: **Overview of VLC: VLM-based concept recognition followed by circuit-based symbolic reasoning.** We use the `pySDD` compiler to compile the symbolic rules into a circuit, SDD in particular. During inference, the VLM is prompted to recognize object concepts in the input image. The generated response is then extracted and processed as the binary inputs to the SDD. The SDD uses compiled rules to execute exact inference over the binary inputs and the outputs are converted to the answer.

for concept recognition and a circuit-based symbolic module to encode the rules (see Fig. 2). Note that this is the simplest neuro-symbolic integration possible, where rules are still explicit (Chen et al., 2025a). While other more sophisticated pipelines and architectures are possible (Manhaeve et al., 2018a; Ahmed et al., 2022; Calanzone et al., 2025; De Smet et al., 2023; Kurscheidt et al., 2025; Lazzari et al., 2026), as we will show in our experiments, this simple strategy suffices when there is supervision over object concepts (Marconato et al., 2023; 2024; 2025).

## 3.1 Phase I: VLM-based Concept Recognition

Leveraging recent advances in visual recognition capabilities of VLMs (Cooper et al., 2025a; Al-Tahan et al., 2024), we directly prompt a VLM to identify object concepts in an image. For each task, we design a specific prompt instructing the VLM to recognize relevant object concepts (*e.g.*, digits, colors, shapes). Taking this prompt and the image as input, the VLM then generates recognition results. To further improve output consistency and facilitate result extraction, we employ in-context learning by incorporating few-shot examples in the prompt. On the MNAdd task, for instance, the VLM takes as input the few-shot examples, an image, and a prompt—*"Image description: The image shows two rows of handwritten digits. Each row represents a multi-digit number.\nQuery: What are the two numbers?"*, and generates a response such as *"Answer: 640, 280"*. Prompts for the other tasks are provided in Appendix B.

## 3.2 Phase II: Circuit-based Symbolic Reasoning

**Symbolic Programs.** In Phase II, we execute the provided rules using a symbolic program rather than relying on a VLM to carry out end-to-end reasoning. A symbolic program is an explicit, executable representation of a rule-based computation, such as a logical formula, a set of constraints, or a small program in a domain-specific language. Given discrete inputs, a symbolic program computes the label by applying the rule exactly. This differs from neural networks, which typically compute outputs from inputs via learned approximations rather than exact rule execution.

**Circuits.** Among many instances of symbolic programs, we focus on Boolean circuits, which are computationally universal and can express any binary function over discrete variables (Arora & Barak, 2009). A Boolean circuit is a directed acyclic graph composed of logical operators (*e.g.*, $\wedge$, $\vee$, and $\neg$) that computes an output from binary inputs. While in knowledge compilation, one often considers circuit classes with certain structural properties, such as decomposability (Darwiche, 2001a), determinism (Darwiche, 2001b),

and smoothness (Darwiche, 2001b), to enable tractable logical inference tasks like weighted model counting (Chavira & Darwiche, 2008), our setting does not strictly rely on them. In our setting, given a discrete assignment to the input variables, the circuit deterministically evaluates the rule and outputs its binary value, *i.e.*, whether the assignment satisfies the rule. This value is then used to derive the answer to the query. We obtain such circuits through knowledge compilation (Darwiche & Marquis, 2002; Chavira & Darwiche, 2005; Darwiche, 2002; Pipatsrisawat & Darwiche, 2008), which converts a symbolic specification of the rule (*e.g.*, a logical formula encoding the reasoning function) into an equivalent graphical structure that supports efficient inference. The circuit representation mechanizes rule execution and can be run on GPUs easily. In our experiments, we compile task-specific rules into circuits using the `pySDD` compiler (Meert & Choi, 2017), which produces sentential decision diagrams (SDDs) (Darwiche, 2011), a class of Boolean circuits in negation normal form (NNF)[2]. Our framework is not tied to SDDs, and other compilers (*e.g.*, those producing d-DNNF circuits) could be substituted without changing the overall procedure.

Unlike many neuro-symbolic approaches (Ahmed et al., 2022; 2023; Chen et al., 2025b; Calanzone et al., 2025), where learning happens by fine-tuning the whole system end-to-end, *i.e.*, by backpropagating through symbolic solvers and circuits (van Krieken et al., 2025), to improve task performance, we opt for the simpler solution of using the circuits as a deterministic mapper in a two-phase architecture in which there is no global fine-tuning. We show empirically that this suffices to achieve performance that is competitive or better than non-neuro-symbolic VLM baselines. Furthermore, we note that we could have used other symbolic formalisms to represent constraints (Michailidis et al., 2025). However, circuits have the benefit to have been extensively used for neuro-symbolic integration, and therefore there is a rich literature and software infrastructure we can exploit (Lazzari et al., 2026).

**Symbolic Reasoning.** Here, we elaborate on how the circuit performs symbolic reasoning over the object concepts recognized by the VLM. Assume we have compiled the task-specific rules into a circuit. As shown in Fig. 2, in Phase I, the VLM is prompted to recognize object concepts from the input image. Thanks to its in-context learning ability, the VLM produces responses that follow the answer format shown in the few-shot examples. This makes it feasible to extract the recognized concepts from the textual response and convert them into binary values that serve as inputs to the circuit. Given this assignment, the circuit deterministically evaluates the rule and produces binary outputs, which we then convert into the final answer to the query.

We illustrate this procedure on the MNAdd task. Phase I produces a response such as "Answer: 640, 280". We parse the two recognized numbers and convert them into their binary representations, 1010000000 and 0100011000, yielding the corresponding input bits to the circuit. This circuit then evaluates the compiled addition rules to produce the output bits, 1110011000, which is converted back to a decimal value 920 as the predicted sum.

## 4 Related Work

**Visual Deductive Reasoning Tasks.** Natural-language deductive reasoning tasks have been widely studied. A variety of benchmarks (Han et al., 2024a;b; Parmar et al., 2024; Tafjord et al., 2021; Tian et al., 2021; Kassner et al., 2021; Saparov et al., 2023) have been established for these tasks, covering diverse types of rules and reasoning patterns, and are used to evaluate the deductive reasoning capabilities of LLMs. Several works (Chen et al., 2024b; Li et al., 2024a; Poesia et al., 2024; Zhang et al., 2023) have further leveraged these benchmarks to investigate specific research questions. In particular, Zhang et al. (2023) use these tasks to examine whether a language model can be trained end-to-end to learn the underlying reasoning function.

However, these deductive reasoning tasks are primarily language-based, with facts, rules, and queries all provided in text. Deductive reasoning tasks with visual inputs are comparatively underexplored, leaving the deductive reasoning capabilities of VLMs less well studied. Existing reasoning benchmarks for VLMs, including those adopted in neuro-symbolic approaches such as Prism and ViperGPT, primarily evaluate performance on general multimodal understanding and reasoning (Chen et al., 2024a), commonsense reasoning (Surís et al., 2023; Gupta & Kembhavi, 2023), abstract reasoning (Zhang et al., 2024b; Hersche et al.,

---

[2]In NNF circuits, NOT gates are restricted to appearing only on input variables.

2024; Yang et al., 2023b), relational or order understanding (Yüksekgönül et al., 2023a; Al-Tahan et al., 2024; Hudson & Manning, 2019), and temporal reasoning (Xiao et al., 2021). The reasoning functions in these tasks tend to be implicit and abstract. In contrast, visual deductive reasoning tasks feature reasoning functions that are explicitly formulated and provided as rules in the input. This explicitness makes it easier to generate datasets with the same reasoning function but varying patterns, providing an effective testbed to study whether a VLM can genuinely learn a reasoning function and thereby achieve robust reasoning.

**VLM-based Paradigms.** Several works (Qiao et al., 2024; Cooper et al., 2025b; Surís et al., 2023; Gupta & Kembhavi, 2023) decompose the end-to-end reasoning process of VLMs to enhance their reasoning capabilities. Specifically, Qiao et al. (2024); Cooper et al. (2025b) decompose the process into two phases—a VLM-based concept recognition phase and an LLM-based reasoning phase—and demonstrate that incorporating an LLM can augment the VLM's external knowledge and reasoning ability. Despite these improvements, the LLM remains a black box, and it is unclear whether it truly encodes a reasoning function. Surís et al. (2023); Gupta & Kembhavi (2023); Kamali & Kordjamshidi (2025) take a different approach by prompting an LLM to generate a code program that decomposes the query into a sequence of steps, which are executed by calling various pretrained models, such as VLMs. These paradigms benefit from introducing intermediate subqueries that are easier to solve; however, they depend heavily on the quality of the generated program, and the reliance on multiple black-box pretrained models increases the risk of propagating errors through incorrect intermediate results.

Beyond reasoning, VLMs have also been used to develop annotation-free concept bottleneck models (CBMs). Traditional CBMs (Koh et al., 2020) consist of a concept recognition model, typically a CNN, and a sparse linear layer. The concept recognition model outputs scores for different concepts in the input image, while the sparse linear layer predicts the final class label based on the concept scores, making the predictions interpretable in terms of the recognized concepts. However, training the concept recognition model requires concept-level annotations, which are often expensive to obtain. Recent approaches (Oikarinen et al., 2023; Yüksekgönül et al., 2023b; Yang et al., 2023a; Debole et al., 2025) eliminate this need by replacing the CNN with a VLM, resulting in annotation-free CBMs. Given a predefined concept vocabulary, with each concept paired with a textual description, the VLM is typically used in one of two ways: either directly prompted to identify whether a concept is present in the image, or used as a feature encoder to generate embeddings for both the image and the concepts, with their cosine similarity serving as concept scores. Overall, in these approaches, the VLM functions as a concept extractor for the downstream classification task.

## 5 Experiments

### 5.1 Experimental Setup

**Datasets.** We perform evaluations on three visual deductive reasoning tasks as described in Section 2. To test robustness under controlled covariate shifts, we use `rsbench` (Bortolotti et al., 2024) to generate, for each task, three datasets that differ only in the number of objects per image. Concretely, we vary this factor from 3 to 5 to 7, yielding the following dataset variants: *MNAdd-3dgt, MNAdd-5dgt, MNAdd-7dgt*; *MNLogic-3dgt*, *MNLogic-5dgt*, *MNLogic-7dgt*; and *KandLogic-3obj, KandLogic-5obj, KandLogic-7obj*. Examples of samples are shown in Appendix D. We also provide evaluations on a different kind of covariate shifts and a more realistic dataset in Appendix F.2 and Appendix F.3, respectively.

**Baselines.** We compare VLC against the following reasoning paradigms. **End2end reasoning**: It refers to the original end-to-end reasoning process of pretrained VLMs. **End2end fine-tuning**: It refers to fine-tuning the pretrained VLMs on datasets with the fewest objects, *i.e.*, MNAdd-3dgt, MNLogic-3dgt, and KandLogic-3obj, respectively. **Prism** (Qiao et al., 2024): Similar to our method, Prism adopts a compositional reasoning paradigm which comprises a concept recognition phase and a reasoning phase. Unlike ours, Prism uses an LLM instead of a symbolic program for the reasoning phase. **ViperGPT** (Surís et al., 2023): ViperGPT prompts a code generation model to produce a program that breaks down a query into sequential steps. A Python interpreter then executes the program by calling different pretrained models, such as detection models or VLMs, to complete each step.

Table 1: **Task accuracy (%) of different reasoning paradigms across covariate shifts on the MNAdd, MNLogic, and KandLogic tasks.** All paradigms adopt a 7B VLM. Results are averaged over 5 random seeds (mean ± standard deviation). The best results are highlighted in bold, and the second-best results are underlined. VLC consistently achieves high performance across covariate shifts, demonstrating its strong reasoning robustness.

| Task | MNAdd | | | MNLogic | | | KandLogic | | |
|---|---|---|---|---|---|---|---|---|---|
| Paradigm | 3dgt | 5dgt | 7dgt | 3dgt | 5dgt | 7dgt | 3obj | 5obj | 7obj |
| End2end RS | 26.99 ± 0.13 | 6.95 ± 0.09 | 1.85 ± 0.02 | 71.73 ± 0.16 | 49.51 ± 0.24 | 50.05 ± 0.02 | 65.05 ± 0.09 | 62.11 ± 0.14 | 66.60 ± 0.10 |
| End2end FT | **79.65 ± 0.04** | 2.53 ± 0.00 | 1.35 ± 0.02 | **99.83 ± 0.00** | 46.55 ± 0.09 | 51.95 ± 0.13 | **99.97 ± 0.00** | 64.28 ± 0.02 | 57.76 ± 0.03 |
| Prism | 50.22 ± 0.13 | 46.21 ± 0.19 | 30.69 ± 0.13 | 51.06 ± 0.18 | 50.19 ± 0.40 | 50.53 ± 0.43 | 58.30 ± 0.10 | 55.06 ± 0.11 | 56.35 ± 0.10 |
| ViperGPT | 19.71 ± 0.04 | 1.92 ± 0.04 | 0.04 ± 0.02 | 85.66 ± 0.11 | 45.36 ± 0.17 | 42.74 ± 0.16 | 96.56 ± 0.08 | **91.18 ± 0.09** | 84.70 ± 0.09 |
| VLC | 54.62 ± 0.06 | **57.79 ± 0.09** | **52.06 ± 0.13** | 97.37 ± 0.04 | **72.78 ± 0.14** | **60.31 ± 0.27** | 95.97 ± 0.07 | 89.97 ± 0.17 | **92.21 ± 0.12** |

**Models.** We adopt the VLM, specifically Qwen2.5-VL-7B-Instruct (Team, 2025), in both our methods and all baselines. In addition, for Prism, we adopt Qwen2.5-7B-Instruct (Team, 2024) as the LLM; for ViperGPT, we employ GPT-4o (Achiam et al., 2023) as the code generation model and GroundingDINO (Liu et al., 2024) as the detection model. Evaluations with a different VLM backbone are provided in Appendix F.1. More implementation details are deferred to Appendix C.

## 5.2 Main Results

**Performance Comparison with Baselines.** Table 1 reports the task accuracy of different reasoning paradigms on the MNAdd, MNLogic, and KandLogic tasks under covariate shifts.

Compared to using pretrained VLMs for end-to-end reasoning, fine-tuning on datasets with the fewest objects (*i.e.*, the 3dgt/obj datasets) significantly improves the in-distribution performance, even achieving near-perfect accuracy on MNLogic and KandLogic tasks. However, these fine-tuned models fail to generalize since their task accuracy on datasets with more objects (*i.e.*, the 5dgt/obj and 7dgt-obj datasets) remains similar to or worse than that of the pretrained models. This suggests that end-to-end fine-tuning may only learn some statistical features from the training data, and thus does not ensure that the VLMs learn the correct reasoning function.

Prism adopts a compositional paradigm, with the reasoning phase delegated to a 7B LLM. We observe that Prism effectively improves performance over end-to-end reasoning on the MNAdd task while degrading accuracy on the MNLogic and KandLogic tasks. These results indicate that the LLM captures the arithmetic reasoning function but struggles with logical reasoning functions. Therefore, delegating reasoning to a black-box LLM cannot guarantee high performance across reasoning tasks, as it depends on the LLM's inherently unknown reasoning capabilities.

ViperGPT generates programs using a code generation model and executes them by calling various pre-trained models. We observe that it attains low accuracy on MNAdd, while demonstrating high accuracy on KandLogic. Given that the generated programs are syntactically and logically correct, this discrepancy in accuracy is likely due to the varying performance of the underlying pretrained models. For instance, as shown in Appendix E, the detection model performs well on KandLogic but fails to detect objects accurately on MNAdd. This underscores a potential limitation of relying on multiple black-box models. Specifically, while the final performance can be high when these models produce correct results, errors at any step may propagate and substantially degrade overall performance.

Finally, we observe that VLC substantially improves the performance over end-to-end reasoning on all datasets across different tasks. On average, they increase accuracy from 54.59% to 82.65% on the 3dgt/obj dataset, from 39.52% to 73.51% on the 5dgt/obj dataset, and from 39.50% to 68.19% on the 7dgt/obj dataset. These consistent gains suggest that **structurally encoding the true reasoning function within a circuit and then incorporating this circuit within the overall architecture enable high-performing and robust reasoning under covariate shifts.**

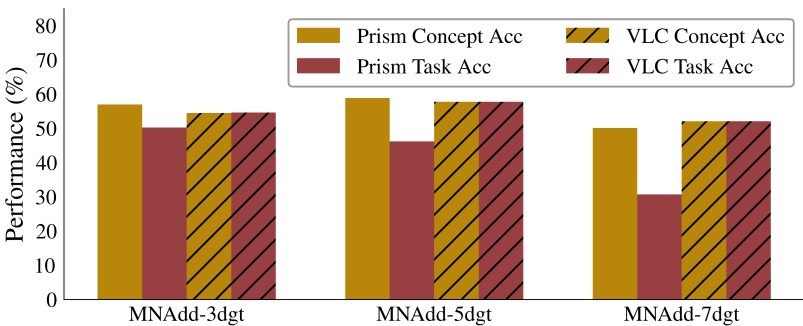

Figure 3: **Concept accuracy and task accuracy of Prism and VLC across covariate shifts on the MNAdd task.** While Prism and VLC achieve similar concept accuracy, VLC maintains a smaller gap between concept accuracy and task accuracy, suggesting that circuit-based reasoning is more robust than LLM-based reasoning in this setting.

**Relationship with Concept Accuracy.** For compositional reasoning paradigms, *i.e.*, Prism and VLC, we further examine the relationship between task accuracy and the accuracy of their VLMs in recognizing object concepts, referred to as *concept accuracy*. Since Prism and VLC adopt the same VLM with the same prompting strategy in their first (concept recognition) phase, they achieve very similar concept accuracy, as shown in Fig. 3. Nevertheless, the task accuracy of Prism is consistently lower than its concept accuracy across datasets, indicating that its reasoning performance is constrained not only by the VLM's recognition capability but also by the LLM's reasoning ability. In contrast, the task accuracy of VLC closely matches its concept accuracy. This suggests that **the performance of VLC on reasoning tasks is dependent solely on the VLM's recognition capability, which is attributed to the fact that VLC structurally encodes the true reasoning function within the circuit.** The ablation study in Appendix G.1 further confirms this. Hence, the incorporation of a circuit effectively improves the reliability of the overall pipeline.

## 5.3 Ablation Studies

**Effect of Fine-tuning on Concept Recognition.** Previous sections have shown that fine-tuning VLMs on reasoning tasks does not guarantee high OOD performance. This raises a natural question: *What is the effect of fine-tuning on recognition tasks?* More specifically, *if we fine-tune VLMs to recognize object concepts in the training data, can the fine-tuned models achieve high recognition performance on OOD data?*

To answer this, we fine-tune VLMs on datasets with the fewest objects, where each sample consists of an image and the concept labels of all objects. We then replace the pretrained VLM in VLC with the fine-tuned one, and evaluate both the concept accuracy and the task accuracy of the resulting model.

As shown in Table 2, **unlike fine-tuning on reasoning tasks, fine-tuning on recognition tasks not only yields high in-distribution concept accuracy but also effectively improves concept accuracy on OOD datasets.** We hypothesize that the features used in concept recognition are relatively stable across distributions, compared with those used in reasoning. Thanks to this improvement, the task accuracy of VLC also increases across all datasets, as it is inherently dependent on the concept accuracy. It turns out that VLC with a fine-tuned VLM becomes the highest-performing model among all reasoning paradigms.

**Effect of Scaling on Reasoning Capabilities of VLMs.** Here, we aim to investigate the impact of model size scaling on the reasoning capabilities of VLMs. To this end, we vary the size of VLMs and evaluate their performance under the end-to-end reasoning paradigm. Fig. 4a presents the task accuracy of this paradigm on the MNAdd-5dgt, MNLogic-5dgt, and KandLogic-5obj datasets. We observe that increasing the model size improves reasoning performance on MNAdd-5dgt and KandLogic-5obj, while failing to improve performance on MNLogic-5dgt, where the task accuracy remains around 50%. These results suggest that **scaling up the size of VLMs may not enhance reasoning for certain reasoning functions**, sharing similar views with Al-Tahan et al. (2024), which shows that scaling cannot aid relational reasoning.

Table 2: **Concept accuracy and task accuracy (%) of VLC before and after concept-recognition fine-tuning across covariate shifts on the MNAdd, MNLogic, and KandLogic tasks.** *"Before ft"* refers to VLC with a pretrained 7B VLM. *"After ft"* denotes VLC using a 7B VLM fine-tuned on the 3dgt/obj datasets for concept recognition. Results are averaged over 5 random seeds (mean ± standard deviation). ↑ means that the corresponding accuracy is improved after fine-tuning. The fine-tuning improves concept recognition on OOD data, which leads to improvements in reasoning on OOD data.

| Task | MNAdd | | | MNLogic | | | KandLogic | | |
|---|---|---|---|---|---|---|---|---|---|
| Metric | 3dgt | 5dgt | 7dgt | 3dgt | 5dgt | 7dgt | 3obj | 5obj | 7obj |
| Concept Acc (before ft) | 54.49 ± 0.05 | 57.76 ± 0.09 | 52.06 ± 0.13 | 97.10 ± 0.03 | 64.50 ± 0.13 | 48.76 ± 0.15 | 91.49 ± 0.06 | 72.97 ± 0.13 | 62.43 ± 0.31 |
| Concept Acc (after ft) | 86.46 ± 0.04 ↑ | 68.32 ± 0.05 ↑ | 53.07 ± 0.10 ↑ | 99.80 ± 0.00 ↑ | 88.39 ± 0.03 ↑ | 74.90 ± 0.08 ↑ | 99.97 ± 0.00 ↑ | 96.50 ± 0.11 ↑ | 85.75 ± 0.15 ↑ |
| Task Acc (before ft) | 54.62 ± 0.06 | 57.79 ± 0.09 | 52.06 ± 0.13 | 97.37 ± 0.04 | 72.78 ± 0.14 | 60.31 ± 0.27 | 95.97 ± 0.07 | 89.97 ± 0.17 | 92.21 ± 0.12 |
| Task Acc (after ft) | 86.46 ± 0.04 ↑ | 68.32 ± 0.05 ↑ | 53.07 ± 0.10 ↑ | 99.80 ± 0.00 ↑ | 90.54 ± 0.04 ↑ | 83.77 ± 0.06 ↑ | 100.00 ± 0.00 ↑ | 99.22 ± 0.05 ↑ | 97.37 ± 0.08 ↑ |

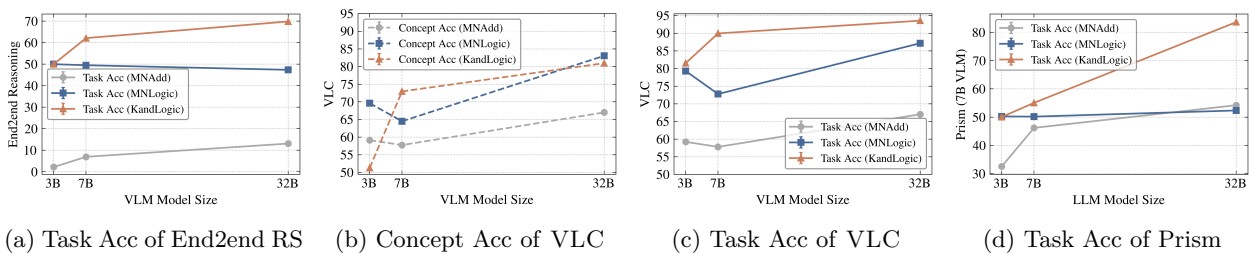

(a) Task Acc of End2end RS    (b) Concept Acc of VLC    (c) Task Acc of VLC    (d) Task Acc of Prism

Figure 4: Performance (%) under model-size scaling. Results are averaged over 5 random seeds, and error bars represent standard deviations. **(a) Effect of scaling up the VLM size in end2end reasoning.** Larger VLMs improve performance on MNAdd and KandLogic, but do not improve performance on MNLogic. **(b-c) Effect of scaling up the VLM size in VLC.** Scaling the VLM to 32B improves both concept accuracy and task accuracy across all three tasks. **(d) Effect of scaling up the LLM size in Prism.** Larger LLMs improve performance on MNAdd and KandLogic, but bring little gain on MNLogic.

**Effect of Scaling on Recognition Capabilities of VLMs.** To explore the impact of model size scaling on the recognition capabilities of VLMs, we apply VLC with varying VLM sizes and examine concept accuracy of the resulting models on the MNAdd-5dgt, MNLogic-5dgt, and KandLogic-5obj datasets. As shown in Fig. 4b, an upward trend in concept accuracy is observed as the model size scales up, despite a slight dip from 3B to 7B. Specifically, when increasing from 3B to 32B, concept accuracy improves by 13%, 19%, and 58% on the three datasets, respectively. These results suggest that **scaling up the size of VLMs effectively enhances their abilities in concept recognition**, aligning with findings from Cherti et al. (2023); Al-Tahan et al. (2024); Zhang et al. (2024a). Besides, due to the compositional nature of VLC, task accuracy also gains improvement, as shown in Fig. 4c.

**Effect of Scaling on Reasoning Capabilities of LLMs.** Here, we attempt to study the effect of model size scaling on the reasoning capabilities of LLMs. To this end, we apply Prism with a fixed 7B VLM and an LLM of varying size, and evaluate the resulting task accuracy on the MNAdd-5dgt, MNLogic-5dgt, and KandLogic-5obj datasets, as shown in Fig. 4d. We find that increasing the LLM size improves reasoning performance on MNAdd-5dgt and KandLogic-5obj, but has little effect on MNLogic-5dgt, where the task accuracy improves by only 2% despite a 29B increase in LLM size. These results suggest that, similar to our findings on VLMs, **scaling up the size of LLMs may not enhance reasoning for certain reasoning functions**.

## 6 Conclusions

In this paper, we study the robustness of the rule-based deductive reasoning of VLMs under controlled covariate shifts and present a neuro-symbolic investigation. We demonstrate that incorporating a symbolic program, a circuit specifically, with a VLM helps improve its performance and robustness on visual deductive

reasoning tasks. In contrast, we find that gradient-based end-to-end fine-tuning and scaling up model size do not guarantee that VLMs capture the correct reasoning function.

We also discuss several limitations and promising directions for future work. First, our current paradigm assumes access to symbolic rules, whereas in practice such rules are often expressed in natural language; extending the paradigm to automatically extract symbolic rules from natural-language descriptions would broaden its applicability. Second, the current symbolic module is task-specific, requiring a separate circuit for each reasoning function; developing a more flexible symbolic module that can accommodate multiple reasoning functions is a promising direction. Third, the overall performance of the proposed paradigm remains dependent on the VLM's recognition capability, suggesting the need for methods that reduce the sensitivity of symbolic reasoning to imperfect visual concept recognition. Finally, our findings are restricted to visual deductive reasoning with explicit task rules and controlled covariate shifts, and do not directly establish robustness in open-ended multimodal reasoning settings. A more detailed discussion of these limitations and directions is provided in Appendix A.

### Acknowledgments

This work is supported by an NSF IIS grant No. 2416897 and a CAREER Award 2442290. AV is supported by the "UNREAL: Unified Reasoning Layer for Trustworthy ML" project (EP/Y023838/1) selected by the ERC and funded by UKRI EPSRC. The views and conclusions expressed in this paper are solely those of the authors and do not necessarily reflect the official policies or positions of the supporting companies and government agencies.

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

## A  Discussions

In this section, we discuss the limitations of the proposed method and provide some potential solutions.

**Extracting Symbolic Rules from Natural Language.**  In this paper, we assume access to symbolic rules for visual deductive reasoning tasks. In practice, however, rules are often expressed in natural language. This raises an important question: *How can symbolic rules be extracted from natural-language descriptions?* We provide a preliminary study in Appendix H that examines the potential of simple prompting-based approaches for extracting symbolic rules from natural language, followed by a discussion of other promising directions.

**Improving Flexibility of the Symbolic Module.**  In this paper, a distinct SDD is constructed for each reasoning function. In other words, switching to a different reasoning function in the visual deductive reasoning task requires replacing the SDD in the symbolic module. This raises a question: *Is it possible to design a more flexible symbolic module that can accommodate multiple reasoning functions?* One promising direction is to design the symbolic module as an ensemble of SDDs, each corresponding to a specific reasoning function. Subsequently, reinforcement learning (RL) (Ouyang et al., 2022) can be employed to adaptively select the appropriate SDD according to the input. The selected SDD is then used to perform the reasoning phase.

**Encoding Rules with Other Symbolic Programs.**  In this paper, we assume that explicit symbolic rules are given for the visual deductive reasoning tasks, and we encode these rules into a circuit. This raises a question: *must the symbolic program be a circuit?* In fact, circuits are not the only symbolic programs capable of encoding such rules. For instance, a code program can also encode a reasoning rule and be executed to derive the answer. In Appendix G.2, we provide a preliminary investigation that replaces the circuit with a code program encoding the same provided rules. We find that the code program yields nearly identical performance to the circuit on the studied datasets. This is expected, since, at the level of rule expressivity, both a code program and a circuit can represent the same deductive rule. These results reinforce our main finding: when the rules are available, exactly encoding them into a symbolic program—whether a circuit or a code program—can improve the robustness of VLMs across covariate shifts.

**Reducing Dependence on Recognition Capabilities of VLMs.**  As shown in our ablation studies, the reasoning performance of VLC is dependent on the recognition capabilities of the VLM. Specifically, strong concept recognition by the VLM leads to high performance of VLC on visual deductive reasoning tasks. Conversely, if the VLM fails to recognize certain object concepts, the reasoning performance of VLC is negatively impacted. To avoid such entanglement, *is it possible to reduce the dependence of* VLC *on the VLM's recognition capabilities?* This also remains an open challenge in the field of neuro-symbolic learning. Addressing this could significantly improve the reliability and robustness of the overall pipeline.

## B  Prompt Design

Fig. 5 presents the prompts designed for the end2end reasoning paradigm on the MNAdd, MNLogic, and KandLogic tasks, respectively. We use them to prompt VLMs to generate the reasoning results.

Fig. 6 presents the prompts designed for VLC on the MNAdd, MNLogic, and KandLogic tasks, respectively. We use them, in the first phase, to prompt VLMs to generate the concept recognition results.

Fig. 7 presents the prompts designed for Prism on the MNAdd, MNLogic, and KandLogic tasks, respectively. We use the prompts shown in the top row, in the first phase, to prompt VLMs to generate the concept recognition results, while using the prompts shown in the bottom row, in the second phase, to prompt LLMs to generate the reasoning results.

For ViperGPT, we use their provided templates to prompt the code generation model, replacing the placeholder string "INSERT_QUERY_HERE" with the prompts shown in Fig. 5.

| MNAdd-3dgt | MNLogic-3dgt | KandLogic-3obj |
|---|---|---|
| ## Image description: The image shows two rows of handwritten digits. Each row represents a 3-digit number. ## Rule: $Y$ is the sum of these two numbers. ## Query: What is $Y$? | ## Image description: The image shows 3 handwritten binary digits. ## Rule: $Y$ is the XOR of these binary digits. ## Query: What is $Y$? | ## Image description: The image shows 3 geometric primitives, each with a specific shape (square, triangle, circle) and color (red, blue, yellow). ## Rule: If all primitives with the same shape have the same color, then $Y$ is True. Otherwise, $Y$ is False. ## Query: What is $Y$? |

Figure 5: **Prompts designed for the end2end reasoning paradigm on different visual deductive reasoning tasks.** These prompts are used to prompt VLMs to generate the reasoning results.

| MNAdd-3dgt | MNLogic-3dgt | KandLogic-3obj |
|---|---|---|
| ## Image description: The image shows two rows of handwritten digits. Each row represents a 3-digit number. ## Query: What are the two numbers? | ## Image description: The image shows 3 handwritten binary digits. ## Query: What are these binary digits? | ## Image description: The image shows 3 geometric primitives, each with a specific shape (square, triangle, circle) and color (red, blue, yellow). ## Query: What are the shape and color of each primitive? |

Figure 6: **Prompts designed for VLC on different visual deductive reasoning tasks.** These prompts are used in the first phase to prompt VLMs to generate the concept recognition results.

| MNAdd-3dgt | MNLogic-3dgt | KandLogic-3obj |
|---|---|---|
| ## Image description: The image shows two rows of handwritten digits. Each row represents a 3-digit number. ## Query: What are the two numbers? | ## Image description: The image shows 3 handwritten binary digits. ## Query: What are these binary digits? | ## Image description: The image shows 3 geometric primitives, each with a specific shape (square, triangle, circle) and color (red, blue, yellow). ## Query: What are the shape and color of each primitive? |
| ## Rule: $Y$ is the sum of these two numbers. ## Query: What is $Y$? Please output the value of $Y$ only. Do not provide explanations. | ## Rule: $Y$ is the XOR of these binary digits. ## Query: What is $Y$? Please output 'True' or 'False' only. Do not provide explanations. | ## Rule: If all primitives with the same shape have the same color, then $Y$ is True. Otherwise, $Y$ is False. ## Query: What is $Y$? Please output 'True' or 'False' only. Do not provide explanations. |

Figure 7: **Prompts designed for Prism on different visual deductive reasoning tasks.** The prompts in the top row are used in the first phase to prompt VLMs to generate the concept recognition results; the prompts in the bottom row are used in the second phase to prompt LLMs to generate the reasoning results.

| **MNAdd-3dgt** |
|---|
| Input variables: a00, a01, a02, a03, a04, a05, a06, a07, a08, a09, b00, b01, b02, b03, b04, b05, b06, b07, b08, b09 
 Constant variables: c00 = False 
 Hidden variables: c01, c02, c03, c04, c05, c06, c07, c08, c09, x00, x01, x02, x03, x04, x05, x06, x07, x08, x09 
 Output variables: Y00, Y01, Y02, Y03, Y04, Y05, Y06, Y07, Y08, Y09, Y10 
 Formulas: x00 = (a00 AND NOTb00) OR (NOTa00 AND b00), Y00 = (x00 AND NOTc00) OR (NOTx00 AND c00), c01 = (a00 AND b00) OR (x00 AND c00), x01 = (a01 AND NOTb01) OR (NOTa01 AND b01), Y01 = (x01 AND NOTc01) OR (NOTx01 AND c01), c02 = (a01 AND b01) OR (x01 AND c01), x02 = (a02 AND NOTb02) OR (NOTa02 AND b02), Y02 = (x02 AND NOTc02) OR (NOTx02 AND c02), c03 = (a02 AND b02) OR (x02 AND c02), x03 = (a03 AND NOTb03) OR (NOTa03 AND b03), Y03 = (x03 AND NOTc03) OR (NOTx03 AND c03), c04 = (a03 AND b03) OR (x03 AND c03), x04 = (a04 AND NOTb04) OR (NOTa04 AND b04), Y04 = (x04 AND NOTc04) OR (NOTx04 AND c04), c05 = (a04 AND b04) OR (x04 AND c04), x05 = (a05 AND NOTb05) OR (NOTa05 AND b05), Y05 = (x05 AND NOTc05) OR (NOTx05 AND c05), c06 = (a05 AND b05) OR (x05 AND c05), x06 = (a06 AND NOTb06) OR (NOTa06 AND b06), Y06 = (x06 AND NOTc06) OR (NOTx06 AND c06), c07 = (a06 AND b06) OR (x06 AND c06), x07 = (a07 AND NOTb07) OR (NOTa07 AND b07), Y07 = (x07 AND NOTc07) OR (NOTx07 AND c07), c08 = (a07 AND b07) OR (x07 AND c07), x08 = (a08 AND NOTb08) OR (NOTa08 AND b08), Y08 = (x08 AND NOTc08) OR (NOTx08 AND c08), c09 = (a08 AND b08) OR (x08 AND c08), x09 = (a09 AND NOTb09) OR (NOTa09 AND b09), Y09 = (x09 AND NOTc09) OR (NOTx09 AND c09), Y10 = (a09 AND b09) OR (x09 AND c09) |
| **MNLogic-3dgt** |
| Input variables: n01, n02, n03 
 Constant variables: None 
 Hidden variables: z01 
 Output variables: Y 
 Formulas: z01 = (n01 AND NOTn02) OR (NOTn01 AND n02), Y = (z01 AND NOTn03) OR (NOTz01 AND n03) |
| **KandLogic-3obj** |
| Input variables: s1_ne_s2, c1_eq_c2, s1_ne_s3, c1_eq_c3, s2_ne_s3, c2_eq_c3 
 Constant variables: None 
 Hidden variables: None 
 Output variables: Y 
 Formulas: Y = (s1_ne_s2 OR c1_eq_c2) AND (s1_ne_s3 OR c1_eq_c3) AND (s2_ne_s3 OR c2_eq_c3) |

Figure 8: Symbolic rules for different visual deductive reasoning tasks.

## C  Implementation Details

Here, we elaborate on the implementation details for all baselines as well as our proposed method. All trainings and evaluations are conducted on a single NVIDIA A100-SXM4 GPU with 80 GB of memory. Specifically, each evaluation under the same setting is repeated five times using random seeds 0, 1, 2, 3, and 4.

**End2end Reasoning.** In the end2end reasoning paradigm, we select Qwen2.5-VL-7B-Instruct as the VLM. We prompt this model using the prompt shown in Fig. 5 along with 5 few-shot examples. The batch size in the data loader is set to 128.

**End2end Fine-tuning.** In the end2end fine-tuning paradigm, we select Qwen2.5-VL-7B-Instruct as the pretrained VLM. We fine-tune the model on the training samples for 5 epochs with a batch size of 4, accumulating gradients over 8 steps. We employ the AdamW optimizer with a fixed learning rate of $2 \times 10^{-4}$, a constant schedule, a warmup ratio of 0.03, and gradient clipping at a maximum norm of 0.3. Evaluation on validation samples is conducted every 100 steps, and the best model, determined by minimum evaluation loss, is restored after training. After fine-tuning, we evaluate the fine-tuned model in the zero-shot setting using the prompt shown in Fig. 5. The batch size in the data loader is set to 128.

**Prism.** In Prism, we select Qwen2.5-VL-7B-Instruct as the VLM and Qwen2.5-7B-Instruct as the LLM. We prompt the VLM using the prompt shown in Fig. 7 (top row) along with 5 few-shot examples, and prompt the LLM using the prompt shown in Fig. 7 (bottom row) in the zero-shot setting. The batch size in the data loader is set to 128.

| image | concept label | label |
|-------|---------------|-------|
| | [809, 489] | 1298 |

| image | concept label | label |
|-------|---------------|-------|
| | [1, 0, 0] | 1 |

| image | concept label | label |
|-------|---------------|-------|
| | [square, blue, triangle, red, circle, blue] | True |

Figure 9: Samples from the MNAdd-3dgt (**top**), MNLogic-3dgt (**middle**), and KandLogic-3obj (**bottom**) datasets.

**ViperGPT.** In ViperGPT, GPT-4o generates a program for each task based on the templates provided in Surís et al. (2023). Once generated, the same program is applied to all test samples for that task. The generated programs frequently invoke both a detection model and a VLM. Specifically, we use Grounding-DINO-Base as the detection model and Qwen2.5-VL-7B-Instruct as the VLM.

**VLC.** The batch size in the data loader of VLC is set to 128. In the first phase of VLC, we prompt Qwen2.5-VL-7B-Instruct using the prompt shown in Fig. 6 along with 5 few-shot examples. In the second phase, the SDD follows the hierarchy of a randomly initialized vtree to compile the symbolic rules shown in Fig. 8.

Across all methods, we use Qwen2.5-VL-7B-Instruct through Hugging Face Transformers with the same default decoding configuration, namely `do_sample=True`, `temperature=1e-6`, and `repetition_penalty=1.05`; unspecified sampling parameters follow Hugging Face Transformers defaults.

## D  Dataset Construction

In this section, we describe the construction of the datasets. Specifically, we use the `rsbench` benchmark to create datasets for various visual deductive reasoning tasks.

**MNAdd.** Datasets for the MNAdd task consist of images depicting two multi-digit numbers, with labels corresponding to the sum of these numbers. Taking the MNAdd-3dgt dataset as an instance, each image contains two rows, where each row comprises three horizontally arranged digit images randomly sampled from MNIST (LeCun et al., 2002). The label is the sum of the two three-digit numbers represented by the rows. We generate a total of 10000 training samples, 3000 test samples, and 2000 validation samples. Sample examples are provided in Fig. 9 (top).

**MNLogic.** Datasets for the MNLogic task consist of images depicting a binary sequence, with labels corresponding to the XOR of the bits. For example, in the MNLogic-3dgt dataset, each image contains three horizontally arranged digit images, each being a 0 or 1, randomly sampled from MNIST (LeCun et al.,

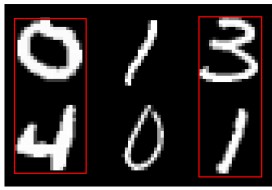 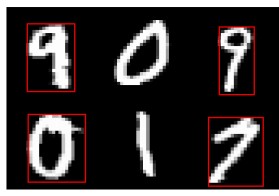 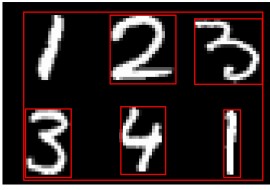

Figure 10: **ViperGPT's detection results on three samples from the MNAdd-3dgt dataset.** The detection model fails to detect each digit correctly.

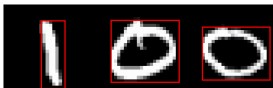 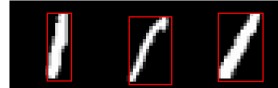 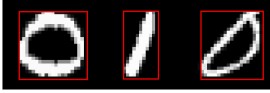

Figure 11: **ViperGPT's detection results on three samples from the MNLogic-3dgt dataset.** The detection model is able to detect each digit correctly.

2002). The label is the XOR of the three bits represented in the image. We generate a total of 10000 training samples, 3000 test samples, and 2000 validation samples. Sample examples are provided in Fig. 9 (middle).

**KandLogic.** Datasets for the KandLogic task consist of images depicting several geometric primitives, with labels indicating the result of a relational check between the shapes and colors of the primitives. Taking the KandLogic-3obj dataset as an example, each image contains three geometric primitives, whose shapes are randomly chosen from {circle, square, triangle} and colors randomly chosen from {red, yellow, blue}. The label indicates whether all primitives with the same shape have the same color. We generate a total of 10000 training samples, 3000 test samples, and 2000 validation samples. Sample examples are provided in Fig. 9 (bottom).

## E    Performance Analysis of ViperGPT

**Effect of the Detection Model.** In the main experiments, we observe that ViperGPT (Surís et al., 2023) achieves competitive performance with the end-to-end reasoning paradigm on the MNLogic and KandLogic tasks, while exhibiting much lower task accuracy on the MNAdd task. Here, we investigate the cause of this inferior performance.

According to the program generated by GPT-4o (Yang et al., 2023b) for solving the MNAdd task (see Fig. 14 (top row)), the first step in ViperGPT's pipeline is to invoke a detection model, GroundingDINO (Liu et al., 2024) in particular, to detect the digits in the input image. As shown in Fig. 10, the detection model fails to correctly detect each digit, leading to errors that propagate through subsequent steps and ultimately result in incorrect answers. In contrast, as shown in Fig. 11 and Fig. 12, the detection model successfully detects

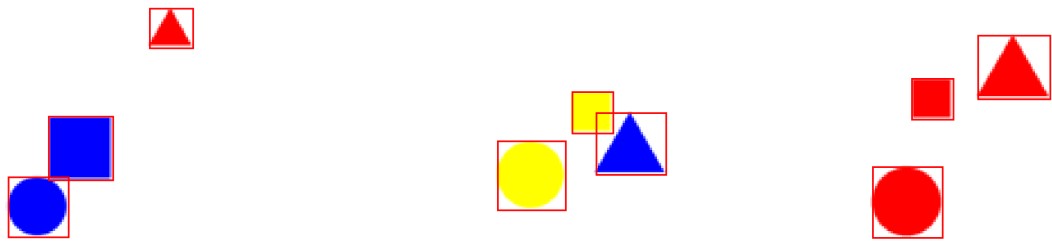

Figure 12: **ViperGPT's detection results on three samples from the KandLogic-3obj dataset.** The detection model is able to detect each geometric primitive correctly.

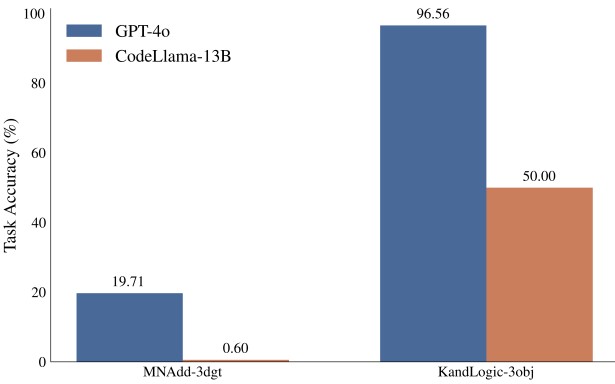

Figure 13: **Task accuracy of ViperGPT using programs generated by GPT-4o and CodeLlama-13B.** CodeLlama-13B produces incorrect programs for MNAdd and KandLogic tasks, resulting in lower task accuracy compared with GPT-4o. These results show that it is important for ViperGPT to select a code generation model that generates syntactically and logically correct programs.

all digits or geometric primitives on the MNLogic and KandLogic tasks, which accounts for ViperGPT's high performance on these tasks.

These findings highlight that ViperGPT's reliance on the detection model enables high task accuracy when detection succeeds but also introduces the risk of cumulative errors that can degrade overall performance.

**Effect of the Code Generation Model.** In ViperGPT, the decomposition of the end-to-end reasoning process is primarily achieved through a program generated by a code generation model. Therefore, the quality of the generated program is of vital importance to the performance of ViperGPT. Here, we investigate the impact of the code generation model on ViperGPT's performance. In the main experiments, we select GPT-4o (Yang et al., 2023b) as the code generation model. Fig. 14 (top row) shows the programs generated by GPT-4o for different tasks. These programs are both syntactically and logically correct, thus enabling effective reasoning by breaking the user query down into a sequence of simpler steps.

We then choose CodeLlama-13B (Roziere et al., 2023) as the code generation model and examine the performance of ViperGPT. Fig. 14 (bottom row) shows the programs generated for different tasks, among which those for the MNAdd and KandLogic tasks are logically incorrect. Thus, the reasoning process is decomposed into incorrect steps. Consequently, as shown in Fig. 13, we observe a significant drop in task accuracy for these tasks. In particular, the incorrect program results in almost 0% accuracy on the MNAdd task.

These findings highlight the importance of selecting a code generation model that produces syntactically and logically correct programs, as errors in this step can critically undermine the overall reasoning process.

# F More Experimental Results

## F.1 Performance with a Different VLM Backbone

In the main experiments, we use Qwen2.5-VL-7B-Instruct as the VLM backbone. To assess the cross-model generalizability of our method, we re-evaluate VLC and all baselines using a different backbone, InternVL3.5-8B-Instruct, with greedy decoding. The results are reported in Table 3; they broadly reproduce the main findings of Section 5.2 and confirm that VLC generalizes across VLM backbones. As in the main paper, the 3dgt/obj setting is in-distribution, while the 5dgt/obj and 7dgt/obj settings are out-of-distribution (OOD).

The end-to-end fine-tuning paradigm effectively improves the performance of the pretrained models on in-distribution data, *i.e.*, the 3dgt/obj datasets. On OOD data, however, the fine-tuned models behave differently across tasks. On KandLogic, fine-tuning substantially improves the task accuracy on both the 5obj and 7obj datasets. On MNAdd, it still improves the OOD task accuracy, though by a smaller margin

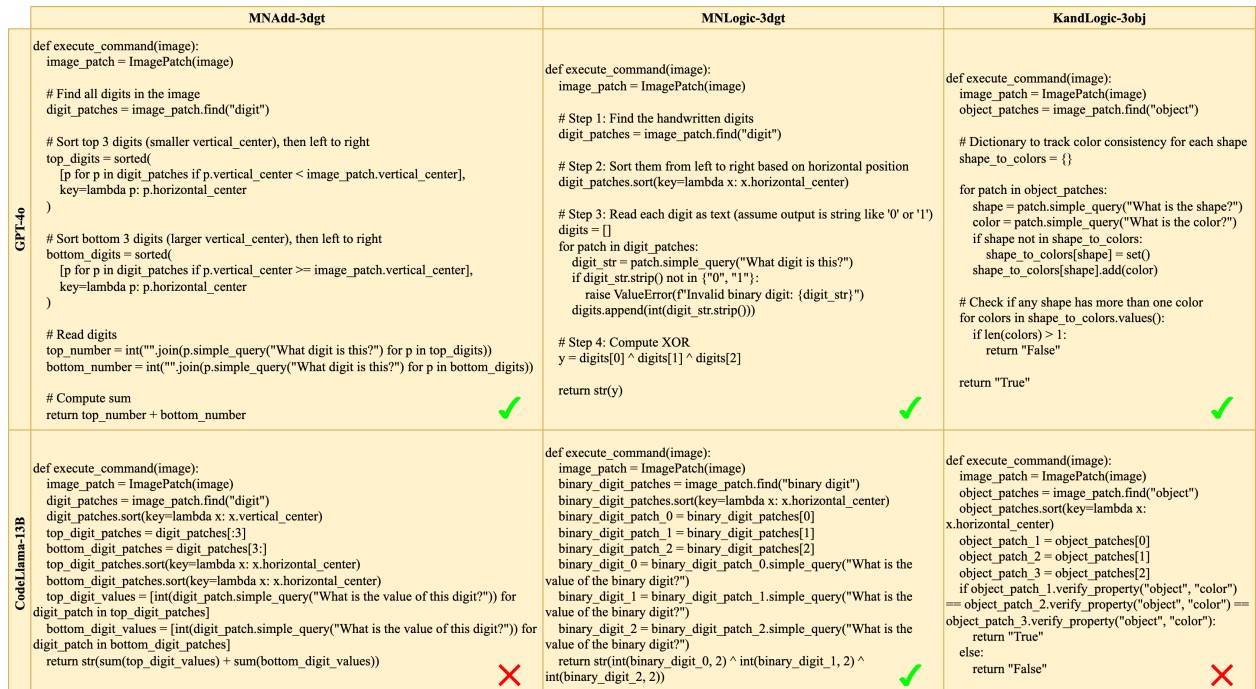

Figure 14: **Programs generated by GPT-4o (top row) and CodeLlama-13B (bottom row) for different visual deductive reasoning tasks.** These programs are used by ViperGPT to decompose the visual deductive reasoning tasks. ✓ and ✗ indicate the correctness of the generated programs.

Table 3: **Task accuracy (%) of different reasoning paradigms across covariate shifts on the MNAdd, MNLogic, and KandLogic tasks, using InternVL3.5-8B-Instruct as the VLM backbone.** The best results are highlighted in bold, and the second-best results are underlined. With the new VLM backbone, VLC still achieves consistently high performance across covariate shifts, demonstrating its strong reasoning robustness.

| Task | MNAdd | | | MNLogic | | | KandLogic | | |
|---|---|---|---|---|---|---|---|---|---|
| Paradigm | 3dgt | 5dgt | 7dgt | 3dgt | 5dgt | 7dgt | 3obj | 5obj | 7obj |
| End2end RS | 0.90 | 0.10 | 0.13 | 43.23 | 53.13 | 50.53 | 59.50 | 78.70 | 51.90 |
| End2end FT | **92.67** | 37.93 | 13.47 | **99.90** | 47.87 | 53.57 | **99.97** | 95.30 | 97.67 |
| Prism | 36.43 | 43.73 | 39.90 | 50.00 | 51.37 | 51.13 | 68.70 | 79.97 | 57.37 |
| ViperGPT | 32.03 | 4.77 | 0.63 | 85.47 | 37.33 | 25.87 | 50.37 | 62.80 | 94.33 |
| VLC | 85.90 | **82.60** | **78.70** | 95.87 | **95.37** | **87.43** | 99.27 | **97.77** | **97.97** |

than on in-distribution data. On MNLogic, in contrast, the task accuracy of the fine-tuned models is nearly identical to that of the pretrained models, indicating no benefit from fine-tuning. Taken together, these results show that the end-to-end fine-tuning paradigm cannot reliably transfer the reasoning ability learned from in-distribution data to OOD data.

The findings for the compositional baselines, Prism and ViperGPT, are also consistent with those in the main paper: each improves over the pretrained models on some tasks but not others. These inconsistencies demonstrate the unreliability of delegating the reasoning phase to black-box models such as LLMs.

Finally, our method, VLC, consistently achieves high performance across all three tasks under covariate shifts: among all approaches, it is the second-best on the in-distribution 3dgt/obj datasets and the best on all the OOD datasets. This consistency implies that the VLM's recognition remains reliable under these

Table 4: **Task accuracy (%) of different reasoning paradigms across new types of covariate shifts on the MNAdd, MNLogic, and KandLogic tasks.** VLC (fine-tuned) refers to our method with a VLM fine-tuned for better concept recognition. All paradigms adopt a 7B VLM. Results are averaged over 5 random seeds (mean ± standard deviation). The best results are highlighted in bold, and the second-best results are underlined. Thanks to the effectiveness of fine-tuning on both in-distribution and OOD concept recognition as well as the circuit's exact encoding of the true reasoning function, VLC achieves the best or second-best task accuracy in all settings after concept-recognition fine-tuning.

| Task | MNAdd | | | MNLogic | | | KandLogic | | |
|---|---|---|---|---|---|---|---|---|---|
| Paradigm | test | color | rotation | test | color | rotation | test | color | shape |
| End2end RS | $26.99 \pm 0.13$ | $24.57 \pm 0.14$ | $13.77 \pm 0.07$ | $71.73 \pm 0.16$ | $68.65 \pm 0.26$ | $67.79 \pm 0.22$ | $65.05 \pm 0.09$ | $61.83 \pm 0.16$ | $61.85 \pm 0.18$ |
| End2end FT | $\underline{79.65 \pm 0.04}$ | $\underline{78.77 \pm 0.07}$ | $\underline{63.57 \pm 0.09}$ | $\mathbf{99.83 \pm 0.00}$ | $\underline{99.43 \pm 0.00}$ | $\mathbf{99.70 \pm 0.00}$ | $\underline{99.97 \pm 0.00}$ | $79.83 \pm 0.02$ | $72.88 \pm 0.04$ |
| Prism | $50.22 \pm 0.13$ | $51.33 \pm 0.18$ | $35.47 \pm 0.23$ | $51.06 \pm 0.18$ | $52.03 \pm 0.22$ | $51.87 \pm 0.17$ | $58.30 \pm 0.10$ | $57.40 \pm 0.18$ | $59.05 \pm 0.30$ |
| ViperGPT | $19.71 \pm 0.04$ | $0.60 \pm 0.00$ | $0.60 \pm 0.00$ | $85.66 \pm 0.11$ | $73.12 \pm 0.02$ | $73.12 \pm 0.02$ | $96.56 \pm 0.08$ | $50.18 \pm 0.02$ | $50.20 \pm 0.00$ |
| VLC | $54.62 \pm 0.06$ | $56.21 \pm 0.14$ | $39.35 \pm 0.04$ | $97.37 \pm 0.04$ | $96.17 \pm 0.06$ | $97.79 \pm 0.04$ | $95.97 \pm 0.07$ | $\underline{91.51 \pm 0.04}$ | $\underline{77.13 \pm 0.11}$ |
| VLC (fine-tuned) | $\mathbf{84.46 \pm 0.04}$ | $\mathbf{84.85 \pm 0.05}$ | $\mathbf{72.45 \pm 0.09}$ | $\underline{99.80 \pm 0.00}$ | $\mathbf{99.67 \pm 0.00}$ | $\underline{99.68 \pm 0.02}$ | $\mathbf{100.00 \pm 0.00}$ | $\mathbf{98.11 \pm 0.04}$ | $\mathbf{80.62 \pm 0.10}$ |

covariate shifts, and that encoding the true reasoning function into a symbolic module is therefore sufficient to ensure reliable and robust overall reasoning.

## F.2 Performance under Additional Covariate Shifts

The covariate shift studied in the main paper varies the number of objects per image while keeping the reasoning function fixed. In this section, we consider additional types of covariate shifts that perturb the visual appearance of objects rather than their count. For each task, we construct two additional OOD variants from the in-distribution test set:

**MNAdd.** Starting from the MNAdd-3dgt test set, we construct two OOD variants: (i) *Color*, where all digits are changed to red; and (ii) *Rotation*, where each digit image is rotated by 10 degrees clockwise.

**MNLogic.** Starting from the MNLogic-3dgt test set, we construct two OOD variants: (i) *Color*, where all digits are changed to red; and (ii) *Rotation*, where each digit image is rotated by 10 degrees clockwise.

**KandLogic.** Starting from the KandLogic-3obj test set, we construct two OOD variants: (i) *Color*, where each object is assigned a color unseen during training; and (ii) *Shape*, where each object is assigned a shape unseen during training.

Table 4 reports the task accuracy of different reasoning paradigms under these additional distribution shifts. The *test* columns report in-distribution accuracy on the original 3dgt/3obj test sets as a reference point, while the remaining columns report accuracy under the corresponding appearance shifts.

The end-to-end fine-tuning paradigm holds up only when the shift is mild. On MNAdd and MNLogic, where recoloring and small rotations leave the digits legible, the fine-tuned VLM remains close to its in-distribution test accuracy. On KandLogic, however, where the OOD variants introduce colors and shapes unseen during training, its accuracy drops from 99.97% to 79.83% (color) and 72.88% (shape).

In contrast, VLC remains comparatively stable across all shifts. On KandLogic in particular, it maintains a smaller gap between in-distribution and OOD accuracy than end-to-end fine-tuning, both before fine-tuning and after it. In fact, across all three tasks and both shift types, VLC with a fine-tuned VLM is the most robust paradigm overall, achieving the best accuracy in seven of the nine settings and the second-best in the other two. Since VLC executes the reasoning function exactly in the circuit and thus depends only on concept recognition, this consistency indicates that recognition features remain relatively stable across the studied covariate shifts.

By comparison, paradigms that delegate reasoning to pretrained black-box models are less reliable: ViperGPT collapses under appearance shifts, falling from 19.71% to 0.60% on MNAdd and from 96.56% to roughly 50% on KandLogic, while Prism remains consistently modest across all settings. These results highlight the unreliability of black-box components for reasoning.

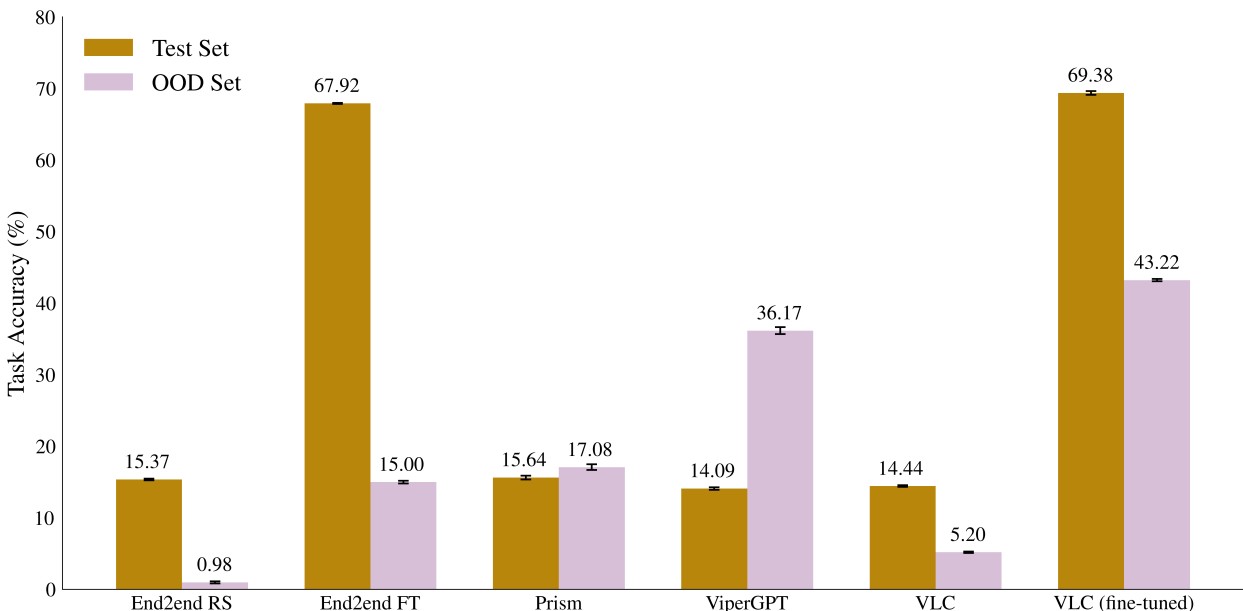

Figure 15: **Task accuracy of different reasoning paradigms on the in-distribution test set and OOD set of the SDD-OIA dataset.** Results are averaged over 5 random seeds, and error bars represent standard deviations. Thanks to the effectiveness of fine-tuning on both in-distribution and OOD concept recognition as well as the circuit's exact encoding of the true reasoning function, VLC achieves the highest task accuracy on both the in-distribution test set and OOD set after concept-recognition fine-tuning.

### F.3 Performance on the SDD-OIA Dataset

In the main paper, we study tasks defined by simple reasoning functions (arithmetic addition, logical XOR, and a relational check). This choice allows us to control generalization in a rigorous way and isolate the effect of covariate shift on the reasoning behavior of each paradigm. In this section, we consider a multi-label autonomous driving task built on a real-world rule set. The goal is to infer which actions out of *forward*, *stop*, *left*, and *right* are safe, depending on the objects (*e.g.*, cars, traffic signs) present in an input dashcam image. The action label is determined by several traffic rules that interact, including relationships between the output actions themselves.

**Dataset.** We use the SDD-OIA (Bortolotti et al., 2024) dataset, in which each image is paired with 4 binary action labels as well as 21 binary object labels. It contains 6,820 training images, 1,464 validation images, 1,464 in-distribution test images, and 1,000 OOD images. The OOD images are generated by holding out a different subset of feasible action configurations during the split construction procedure.

**Rules.** The rules encode the common traffic rules used in the real world. Based on a set of binary concepts indicating the presence of different obstacles on the road, these rules specify the conditions under which the vehicle may move *forward* (`green_light`$\lor$`follow`$\lor$`clear` $\Rightarrow$ `forward`), must *stop* (`red_light`$\lor$`stop_sign`$\lor$ `obstacle` $\Rightarrow$ `stop`), and may turn *left* and *right*, as well as logical relationships between actions (*e.g.*, `stop` $\Rightarrow \neg$ `forward`).

**Evaluation metric.** Task accuracy is defined such that a sample is counted as correct only when all four actions are predicted correctly.

**Results.** The performance of different reasoning paradigms is reported in Fig. 15. The results show that the end-to-end fine-tuning paradigm suffers from a substantial generalization gap between the test and OOD sets, suggesting that it may learn incorrect or incomplete rules. By contrast, the compositional paradigms

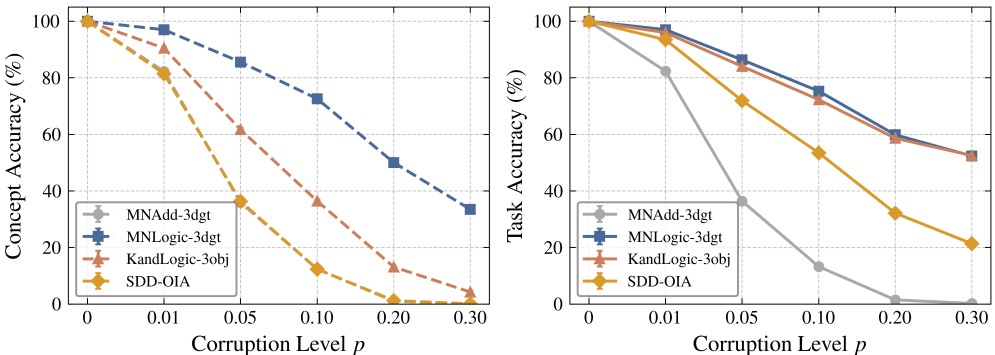

Figure 16: **Concept accuracy (left) and task accuracy (right) of VLC under synthetic concept corruption at increasing noise levels** $p$. Results are averaged over 5 repetitions, and error bars represent standard deviations. As $p$ increases, concept accuracy drops, and task accuracy drops accordingly, confirming that the downstream performance of VLC is governed by the recognition quality of its neural module.

are relatively more stable under covariate shift. For instance, Prism obtains 15.64% task accuracy on the test set and 17.08% on the OOD set.

The results are also consistent with our ablation findings in Section 5.3—fine-tuning effectively improves concept recognition performance on both the test and OOD sets. Specifically, after fine-tuning on the concept labels in the training data, the concept accuracy of the VLM increases from 0% to 37.33% on the test set and from 0% to 31.92% on the OOD set. Benefiting from this, our method with a fine-tuned VLM achieves the highest task accuracy.

Overall, these experimental results on a task with real-world rules suggest that the end-to-end fine-tuning paradigm struggles to learn the underlying prediction rules from data and therefore may not effectively improve OOD reasoning performance; nevertheless, fine-tuning remains useful for OOD concept recognition, and neuro-symbolic paradigms would benefit from this.

# G   More Ablation Studies

## G.1   Error Propagation from Perception to Reasoning

Because VLC performs VLM-based concept recognition followed by exact symbolic reasoning, its overall performance is coupled to concept-recognition accuracy. Section 5.2 already provides partial empirical evidence for this. Here, we add a controlled error-propagation analysis that quantifies how task accuracy degrades as concept-recognition errors increase.

To this end, we inject synthetic noise into the ground-truth concept labels at different corruption levels and feed the corrupted concepts into the same compiled circuit, measuring the resulting degradation in task accuracy. Specifically, for each test sample, we start from the ground-truth concept vector $c$. For each noise level $p$, we repeat the following procedure five times: independently flip each concept with probability $p$ to obtain a corrupted concept vector $\tilde{c}$, feed $\tilde{c}$ into the compiled circuit to obtain the prediction $\hat{y}$, and compute both concept accuracy and task accuracy. Fig. 16 reports results on the MNAdd-3dgt, MNLogic-3dgt, KandLogic-3obj, and SDD-OIA datasets.

When no noise is injected ($p = 0$), both concept accuracy and task accuracy are 100%. This is expected because VLC's symbolic module compiles the true reasoning function. As $p$ increases, concept accuracy drops, and task accuracy drops accordingly, confirming that the downstream performance of VLC is governed by the recognition quality of its neural module. Moreover, because the circuit executes the true reasoning function exactly, any sample whose concept vector is entirely correct also yields a correct prediction; task accuracy is therefore lower-bounded by concept accuracy at every noise level, as the figures confirm. Overall,

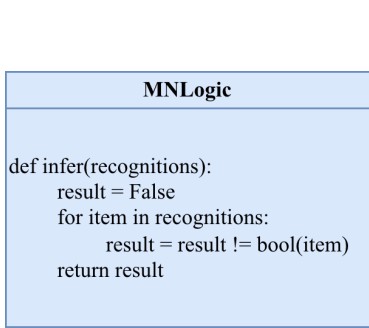

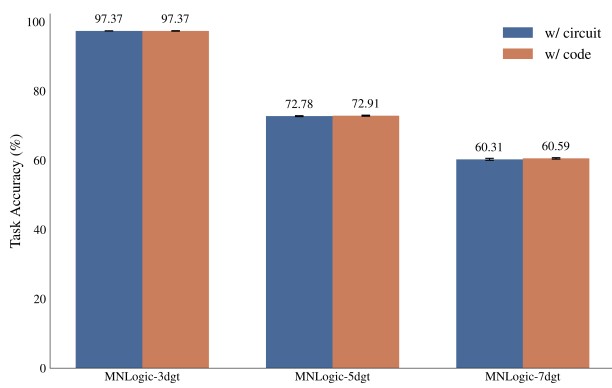

Figure 17: Comparison between circuit- and code-based symbolic modules. **Left: the Python program used to implement the MNLogic reasoning function. Right: task accuracy when the symbolic module is implemented as an SDD circuit or as Python code.** The two symbolic modules achieve nearly identical performance, since both can exactly encode the true reasoning function.

this controlled analysis shows that the symbolic module itself introduces no additional approximation error, and that the degradation in final task performance is driven by errors in concept recognition.

### G.2 Encoding Rules with Other Symbolic Programs

In this section, we conduct a preliminary study of whether symbolic programs other than SDDs can encode reasoning rules. We focus on executable Python programs as a simple alternative. Specifically, for the MNLogic task, we translate the reasoning rules in Eq. (2) into the Python program shown in Fig. 17 (left). At inference time, the VLM first recognizes the digits in the input image. The recognized digits are then passed to a Python interpreter, which executes the program and returns the final answer to the user query.

The results are shown in Fig. 17 (right). Using the Python program as the symbolic module leads to nearly identical performance to using an SDD. This is expected, since, at the level of rule expressivity, both a code program and a circuit can represent the same deductive rule. These results further support our main finding: when the reasoning rules are known, exactly encoding them into a symbolic module, whether as a circuit or as a code program, can improve the robustness of VLMs under covariate shifts.

## H Inferring Symbolic Rules from Natural-Language Descriptions

In open-domain scenarios, the reasoning rules are rarely provided in the symbolic form; automatically extracting symbolic rules from natural-language (NL) descriptions is an important but challenging open problem. While the main goal of this paper is to show that encoding symbolic rules into a circuit can reliably improve reasoning robustness, here we take an initial step toward this direction and probe whether a simple prompting-based approach can recover the symbolic rule from an NL description.

We consider zero-shot prompting, where we directly prompt an LLM to generate Boolean formulas from NL descriptions. For each task, we use the NL description from the main paper together with four paraphrased versions, resulting in five NL descriptions in total, and prompt the LLM (Qwen2.5-7B-Instruct) separately with each. We evaluate the generated rules with two metrics: (i) the *validity percentage*, the percentage of generated outputs that can be successfully compiled into an SDD; and (ii) the *truth table alignment*, the percentage of input assignments for which a valid generated rule matches the ground-truth symbolic rule.

Table 5 summarizes the results, from which we observe three main trends. First, for relatively simple tasks such as MNLogic-3dgt, the LLM can infer symbolic rules of high quality, achieving 80% truth table alignment. Second, for more complex tasks such as KandLogic-3obj and SDD-OIA, the validity percentage or the truth table alignment drops substantially. Third, for tasks such as MNAdd-3dgt, which require the model to draw on its internal knowledge (*e.g.*, full-adder logic), performance becomes extremely poor.

Table 5: **Performance of zero-shot prompting for symbolic rule inference from NL descriptions.** The results show that it is challenging to generate the exact symbolic rules using zero-shot prompting.

| Dataset | Validity (%) | Truth Table Alignment (%) |
|---|---|---|
| MNAdd-3dgt | 40.00 | 0.49 |
| MNLogic-3dgt | 100.00 | 80.00 |
| KandLogic-3obj | 80.00 | 45.31 |
| SDD-OIA | 100.00 | 50.70 |

Table 6: Inference time (in seconds) of different reasoning paradigms on the MNAdd-3dgt, MNLogic-3dgt, and KandLogic-3dgt test sets.

| Paradigm | MNAdd-3dgt | MNLogic-3dgt | KandLogic-3obj |
|---|---|---|---|
| End2end RS | 207.67 | 211.28 | 432.43 |
| End2end FT | 40.34 | 38.24 | 50.02 |
| Prism | 250.92 | 238.59 | 522.66 |
| ViperGPT | 2106.14 | 1274.43 | 3605.11 |
| VLC | 201.52 | 198.39 | 451.64 |

Beyond this simple zero-shot prompting strategy, further strategies such as prompt engineering can be studied. More advanced approaches may also help: for example, recent work (Yang et al., 2024) suggests that targeted training data and fine-tuning strategies (*e.g.*, RLHF) can effectively improve a model's ability to infer symbolic rules from NL descriptions. Overall, our preliminary results suggest that generating the exact symbolic rules remains challenging. We therefore view automatic rule acquisition as an important research direction that warrants further study.

## I Inference Time Analysis

Beyond task accuracy, the practical trade-offs between reasoning paradigms also depend on their computational cost. To provide a better understanding of these trade-offs, we report the wall-clock inference time of each paradigm. Table 6 reports the inference time of each reasoning paradigm on the MNAdd-3dgt, MNLogic-3dgt, and KandLogic-3obj test sets.

We observe that the end-to-end fine-tuned VLM (End2end FT) is the fastest on all three datasets. This is because, after fine-tuning, the model can directly generate outputs in the desired format and therefore does not require few-shot examples in the prompt. Compared with it, the pretrained VLM (End2end RS) and VLC have similar inference times, as they both rely on five few-shot examples in the prompt: the former uses them to demonstrate the reasoning output format, while the latter uses them to demonstrate the concept recognition output format. The longer inference time of Prism than VLC suggests that, on these studied datasets, LLM-based reasoning is slower than circuit-based reasoning. ViperGPT is by far the slowest paradigm because it may invoke various pretrained models multiple times for each sample during inference, which leads to substantially higher runtime.

Overall, these results reveal a trade-off between flexibility and efficiency: paradigms with greater flexibility tend to be less efficient. For example, compared with Prism, ViperGPT is more flexible because it can invoke a wider range of pretrained models many times, but it uses higher runtime. Compared with VLC, Prism is more flexible because it relies on the general reasoning ability of an LLM rather than strictly following predefined symbolic rules, while it uses higher runtime.

