# OpenReview forum: "Can VLMs Reason Robustly? A Neuro-Symbolic Investigation"
_TMLR — Accepted by TMLR_

### Review · Reviewer_mBuh · 2026-04-06

**Summary Of Contributions:**

**Technical:** Current reasoning pipeline use VLMs for both recognition and reasoning. This submission claim to separate this process: Only using VLMs for recognition, and then using explicit rule-based operations (e.g., Circuit-based Symbolic Rules) for reasoning.

**Experimental:** They evaluate the proposed model via comparisons with existing VLM paradigms and ablation studies on simple MNIST hand written dataset and three simple logic rules cases.

**Audience:**

Yes

**Audience Explanation:**

The topic is interesting and relevant to several parts of the TMLR audience, especially researchers working on VLM reasoning. The paper discussed whether current VLMs actually learn a reusable reasoning function under distribution shift. Even though the evaluated setting is narrow, the results provide a concrete case where a fully implicit recognition-plus-reasoning pipeline underperforms a decoupled pipeline with explicit symbolic deduction.

**Broader Impact Concerns:**

No concerns for this part.

**Claims And Evidence:**

No

**Claims Explanation:**

1. The empirical evidence is not convincing enough to support broader claims about robust VLM reasoning. The experiments are conducted on only three relatively simple visual deductive reasoning tasks generated from RSBench: arithmetic addition over digits, XOR over binary digits. The symbolic rules considered here are low-complexity: addition, XOR, and a pairwise relational consistency rule, and does not demonstrate advantages under more complex compositional rules.

2. The method assumes that the clear rule formulation is already given in a symbolic form suitable for compilation. In practice, this is often the hardest part of the problem. Many realistic reasoning tasks are open-domain and under-specified, and the challenge is not only executing a rule but also formalizing the relevant rule from natural language or context. The paper briefly acknowledges this limitation and lists automatic rule extraction from natural language as future work, but this gap may limit the applicability of the current evidence.

**Requested Changes:**

1. Demonstrate the method under more complex rules. The current rules: addition, XOR, and a relational consistency check, are simple and particularly amenable to exact circuit execution. The paper would be stronger if it included experiments with more compositional and structurally challenging rules

2. Discuss more about rule formulation itself. A central practical issue is that in open-domain scenarios the rule is rarely given in a directly compilable symbolic form. The paper briefly mentions automatic extraction of symbolic rules from natural language as future work, but this point deserves more substantial treatment because it is one of the main obstacles to real-world applicability.

---

> ### Author Response · Authors · 2026-04-22
> **Response to Q1**
>
> We sincerely thank the reviewer for the constructive comments! We're glad that you found our paper interesting to the VLM reasoning community and acknowledged the simple yet concrete case we provided to support our claims!
>
> > **Q1:** Demonstrate the method under more complex rules. The current rules are simple. The paper would be stronger if it included experiments with more compositional and structurally challenging rules.
>
> **A1:** First, we'd like to clarify that, in our paper, we focus on relatively simple rules given the following concerns:
>
> - Generalization can be controlled in a rigorous way.
> - We want to show that even under these simple rules, end-to-end fine-tuned VLMs fail to generalize to OOD samples, thus suggesting that end-to-end fine-tuning does not reliably learn the underlying reasoning rules from data.
>
> However, we agree with the reviewer that including experiments with more complex rules can better support our claims. To this end, we consider an additional task with real-world rules. The experimental settings are listed below:
>
> - **Task:** We adopt a multi-label autonomous driving task for studying visual reasoning in high-stakes scenarios. The goal is to infer what actions out of *forward, stop, left, right* are safe depending on what objects (*e.g.*, cars, traffic signs) are present in an input dashcam image.
> - **Dataset:** We use the SDD-OIA [1] dataset, in which each image is paired with 4 binary action labels as well as 21 binary object labels. It contains 6,820 training images, 1,464 validation images, 1,464 in-distribution test images, and 1,000 OOD images. The OOD images are generated by holding out a different subset of feasible action configurations during the split construction procedure.
> - **Rules:** The rules describe the common traffic rules used in the real world. Specifically, based on a set of binary concepts indicating the presence of different obstacles on the road, these rules specify conditions for being able to *forward* (green_light $\vee$ follow $\vee$ clear $\Rightarrow$ forward), *stop* (red_light $\vee$ stop_sign $\vee$ obstacle $\Rightarrow$ stop), and for turning *left* and *right*, as well as relationships between actions (like stop $\Rightarrow$ $\neg$forward).
> - **Evaluation metric:** Task accuracy is defined such that a sample is counted as correct only when all 4 actions are predicted correctly. Concept accuracy is defined such that a sample is counted as correct only when all 21 objects are predicted correctly.
>
> The performance of our method and baselines is reported in **Table 1**.
>
> **Table 1:** Task accuracy of different reasoning paradigms on the SDD-OIA task under covariate shifts. VLC (fine-tuned) refers to our method with a VLM fine-tuned for better concept recognition. All paradigms adopt a 7B VLM. Results are averaged over 5 random seeds (mean $\pm$ standard deviation). The best results are highlighted in bold, and the second-best results are underlined.
>
> | Paradigm         | Task Acc on Test Set (%) | Task Acc on OOD Set (%) |
> | ---------------- | ------------------------ | ----------------------- |
> | End2end RS       | 15.37 $\pm$ 0.12         | 0.98 $\pm$ 0.15         |
> | End2end FT       | $\underline{67.92 \pm 0.08}$  | 15.00 $\pm$ 0.20        |
> | Prism            | 15.64 $\pm$ 0.27         | 17.08 $\pm$ 0.40        |
> | ViperGPT         | 14.09 $\pm$ 0.17         | $\underline{36.17 \pm 0.50}$ |
> | VLC              | 14.44 $\pm$ 0.12         | 5.20 $\pm$ 0.10         |
> | VLC (fine-tuned) | **69.38 $\pm$ 0.26**     | **43.22 $\pm$ 0.16**    |
>
> The results show that the end-to-end fine-tuning paradigm suffers from a substantial generalization gap between the test and OOD sets, suggesting that it may learn incorrect or incomplete rules. By contrast, the compositional paradigms are relatively more stable under covariate shift. For instance, Prism obtains 15.64% task accuracy on the test set and 17.08% on the OOD set.
>
> The results are also consistent with our ablation findings—fine-tuning effectively improves concept recognition performance on both the test and OOD sets. Specifically, after fine-tuning on the concept labels in the training data, the concept accuracy of the VLM increases from 0% to 37.33% on the test set and from 0% to 31.92% on the OOD set. Benefiting from this, our method with a fine-tuned VLM achieves the highest task accuracy.
>
> Overall, **these experimental results on a task with real-world rules suggest that the end-to-end fine-tuning paradigm struggles to learn the underlying prediction rules from data and therefore may not effectively improve OOD reasoning performance; nevertheless, fine-tuning remains useful for OOD concept recognition, and neuro-symbolic paradigms would benefit from this.**
>
> [1] Bortolotti, Samuele, et al. "A neuro-symbolic benchmark suite for concept quality and reasoning shortcuts." *Advances in neural information processing systems* 37 (2024): 115861-115905.

---

> ### Author Response · Authors · 2026-04-22
> **Response to Q2 (part1)**
>
> > **Q2:** Discuss more about the rule formulation itself. In open-domain scenarios, the rule is rarely given in a symbolic form. The paper briefly mentions automatic extraction of symbolic rules from natural language as future work, but this point deserves more substantial treatment because it is one of the main obstacles to real-world applicability.
>
> **R2:** We agree that extracting symbolic rules from natural-language (NL) descriptions is an important but challenging open problem. While the main goal of our paper is to show that encoding symbolic rules into a circuit can reliably improve reasoning robustness, here we’d like to take an initial step to explore how simple prompting-based approaches may help and then discuss other promising directions. The experimental setup for these approaches is listed below.
>
> - **Strategies.** We consider four prompting strategies: **(i)** zero-shot prompting, where we directly prompt an LLM to generate a Boolean formula from the NL description; **(ii)** few-shot prompting, where we additionally provide one example; **(iii)** zero-shot chain-of-thought (CoT) prompting, where we ask the LLM to reason through useful intermediate Boolean conditions and introduce intermediate variables when helpful; and **(iv)** few-shot CoT prompting.
> - **Method.** For each prompting strategy, we use not only the canonical NL description from our paper but also four paraphrased versions, resulting in five NL descriptions in total. We then prompt the LLM (specifically, Qwen2.5-7B-Instruct) separately with each of the five descriptions.
> - **Evaluation metrics.** We use two metrics: **(i)** validity percentage, the percentage of generated outputs that can be successfully compiled into an SDD; and **(ii)** truth table alignment, for valid outputs, the percentage of input assignments for which the generated rule matches the ground-truth symbolic rule.
>
> **Table 2:** Performance of different prompting strategies for symbolic rule inference from NL descriptions.
>
> |                | Zero-Shot Prompting |                       | Few-Shot Prompting  |                       | Zero-Shot CoT Prompting |                       | Few-Shot CoT Prompting |                       | Average             |                       |
> | -------------- | ------------------- | --------------------- | ------------------- | --------------------- | ----------------------- | --------------------- | ---------------------- | --------------------- | ------------------- | --------------------- |
> | Dataset        | Validity Percentage | Truth Table Alignment | Validity Percentage | Truth Table Alignment | Validity Percentage     | Truth Table Alignment | Validity Percentage    | Truth Table Alignment | Validity Percentage | Truth Table Alignment |
> | MNAdd-3dgt     | 40.00%              | 0.49%                 | 80.00%              | 0.01%                 | 80.00%                  | 5.69%                 | 60.00%                 | 0.00%                 | 65.00%              | 1.55%                 |
> | MNLogic-3dgt   | 100.00%             | 80.00%                | 80.00%              | 75.00%                | 100.00%                 | 47.50%                | 100.00%                | 85.00%                | 95.00%              | 71.88%                |
> | KandLogic-3obj | 80.00%              | 45.31%                | 100.00%             | 59.38%                | 100.00%                 | 44.38%                | 100.00%                | 59.38%                | 95.00%              | 52.11%                |
> | SDD-OIA        | 100.00%             | 50.70%                | 100.00%             | 39.48%                | 80.00%                  | 19.86%                | 100.00%                | 37.33%                | 95.00%              | 36.84%                |

---

> ### Author Response · Authors · 2026-04-22
> **Response to Q2 (part 2)**
>
> **Table 2** summarizes the results. From the zero-shot prompting results, we observe three main trends. **(a)** For relatively simple tasks such as MNLogic-3dgt, the LLM can infer symbolic rules with high quality, achieving 80% truth table alignment. **(b)** For more complex tasks, such as KandLogic-3obj and SDD-OIA, the validity percentage or truth table alignment drops substantially. **(c)** For tasks such as MNAdd-3dgt, which require the model to utilize its internal knowledge (e.g., full adder logic), performance becomes extremely poor.
>
> The other prompting strategies help to different extents, depending on the task. For example, few-shot prompting improves performance on KandLogic-3obj, zero-shot CoT helps on MNAdd-3dgt, and few-shot CoT gives the best results on MNLogic-3dgt.
>
> More advanced approaches may improve this further. For example, recent work [2] suggests that targeted training datasets and fine-tuning strategies (e.g., RLHF) can effectively improve a model's ability to infer symbolic structure from NL descriptions.
>
> Overall, these results suggest that **appropriate prompting often enables the LLM to produce a valid symbolic rule, but generating the correct rule remains much more difficult.** We therefore view automatic rule acquisition as an important and largely orthogonal research direction that requires further study. We will expand the above discussion in the revised paper, making it clearer in the limitations and future work sections.
>
> [2] Yang, Yuan, et al. "Harnessing the power of large language models for natural language to first-order logic translation." *Proceedings of the 62nd Annual Meeting of the Association for Computational Linguistics (Volume 1: Long Papers)*. 2024.
>
> Once again, thanks for your constructive comments, which motivate us to develop more supportive evidence in practical scenarios!

---

### Review · Reviewer_vW7D · 2026-04-09

**Summary Of Contributions:**

This paper investigates the robustness of vision-language models (VLMs) under controlled covariate shifts in visual deductive reasoning tasks. The authors show empirically that end-to-end fine-tuning can achieve high in-distribution accuracy but fails to generalize when the number of objects changes while the underlying reasoning rule remains the same. To address this limitation, the paper proposes a simple neuro-symbolic framework (VLC) that separates perception from reasoning by using a VLM for concept recognition and a circuit-based symbolic module to execute the reasoning rules deterministically.

Key strengths of the paper include a clean and well-controlled evaluation setup for studying reasoning under distribution shift, clear empirical evidence highlighting limitations of end-to-end reasoning, and a simple yet interpretable neuro-symbolic design that demonstrates consistent robustness improvements across tasks.

Key weaknesses include the reliance on synthetic tasks with explicitly provided rules, which may limit the generalizability of the conclusions to more realistic reasoning settings, and the relatively narrow scope of robustness evaluation (primarily object-count variation).

**Audience:**

Yes

**Audience Explanation:**

Researchers working on vision-language models, robustness, and neuro-symbolic reasoning would likely find the findings informative, particularly the controlled analysis of reasoning under distribution shifts.

**Claims And Evidence:**

No

**Claims Explanation:**

Partially. The empirical evidence is generally clear and well-controlled, and the results convincingly demonstrate improved robustness within the specific task setting considered. However, the scope of the evaluation is relatively limited to synthetic rule-based tasks under controlled covariate shifts, which makes it less clear whether the conclusions generalize to more realistic and diverse reasoning scenarios.

**Requested Changes:**

1. Clarify the scope and definition of “reasoning robustness.” The paper evaluates robustness primarily through controlled covariate shifts in synthetic visual deductive reasoning tasks (e.g., addition, XOR, relational consistency). While this setup is clean and well-controlled, the manuscript currently makes relatively broad claims about “robust reasoning” in VLMs. I recommend explicitly clarifying that the conclusions are restricted to rule-based deductive reasoning under controlled distribution shifts, rather than general reasoning in open-ended multimodal settings. This clarification is important to ensure that the claims are properly scoped and accurately interpreted by readers.

2. Provide stronger justification for the practical relevance of the task setting. The current framework assumes that symbolic reasoning rules are explicitly available and can be directly compiled into circuits. While this enables a clean evaluation of reasoning robustness, this assumption may limit the applicability of the proposed approach to real-world multimodal reasoning scenarios, where rules are often implicit, incomplete, or expressed in natural language. To strengthen the practical relevance of the work, I encourage the authors to:
- Clarify realistic application domains where explicit symbolic rules are naturally available and reliable.
- Discuss the gap between the current experimental setting and real-world reasoning tasks, where rules typically need to be inferred rather than provided.
- Outline potential directions for extending the framework to handle natural-language or partially specified rules.

3. Strengthen the empirical analysis of generalization beyond object-count shifts. The current evaluation focuses on increasing the number of objects per image while keeping the reasoning function fixed. This is a meaningful form of covariate shift, but it represents only one axis of distribution shift. To strengthen the robustness claims, the authors should either:
- include at least one additional shift type (e.g., visual appearance, object style, noise, occlusion, or concept ambiguity), or
- provide a clear justification for why object-count variation is a sufficiently representative robustness stress test.

4. Add an explicit analysis of error propagation from perception to reasoning. Since the proposed VLC pipeline relies on VLM-based concept recognition followed by deterministic symbolic reasoning, overall performance is tightly coupled to recognition accuracy.
A short analysis quantifying how reasoning accuracy degrades under controlled recognition errors (e.g., simulated noise in concept predictions) would improve understanding of system robustness and practical deployment behavior.

---

> ### Author Response · Authors · 2026-04-22
> **Responses to Q1 and Q2 (part1)**
>
> We sincerely thank the reviewer for the thoughtful feedback! We are glad that the reviewer found our paper to present a simple yet interpretable neuro-symbolic design, a clean and well-controlled evaluation setup, and clear empirical evidence.
>
> > **Q1:** Clarify the scope and definition of reasoning robustness. I recommend explicitly clarifying that the conclusions are restricted to rule-based deductive reasoning under controlled distribution shifts, rather than general reasoning in open-ended multimodal settings. This clarification is important to ensure that the claims are properly scoped and accurately interpreted by readers.
>
> **A1:** Thanks for the helpful suggestion. We will refine the statements in the paper to make sure the claims are properly scoped and accurately interpreted by readers. In particular,
>
> - Abstract. We will revise "Experiments on three visual deductive reasoning tasks with distinct rule sets show that VLC consistently achieves strong performance under covariate shifts, highlighting its ability to support robust reasoning." **to** "Experiments on three *simple* visual deductive reasoning tasks with distinct rule sets show that VLC consistently achieves higher task accuracy on out-of-distribution data than other reasoning paradigms."
> - Section 1 (Introduction). We will add a footnote to clarify the reasoning robustness in our context: "Throughout this paper, we use reasoning robustness to refer specifically to the ability to apply a fixed, explicitly provided deductive rule under controlled covariate shifts in perceptual input, rather than to general-purpose reasoning in open-ended multimodal settings."
> - Section 6 (Conclusions).
>   - We will revise "In this paper, we study the problem of robust VLM reasoning and present a neuro-symbolic investigation." **to** "In this paper, we study the robustness of rule-based deductive reasoning in VLMs under controlled covariate shifts and present a neuro-symbolic investigation."
>   - We will also clarify in the limitation paragraph that our conclusions are restricted to visual deductive reasoning with explicit task rules and controlled covariate shifts, and do not directly establish robustness in open-ended multimodal reasoning settings.
>
> We hope these revisions could make the scope of our claims clearer and help readers interpret our contributions more accurately.
>
> > **Q2:** Provide a stronger justification for the practical relevance of the task setting. The current framework assumes that symbolic reasoning rules are explicitly available and can be directly compiled into circuits. While this enables a clean evaluation of reasoning robustness, this assumption may limit the applicability of the proposed approach to real-world multimodal reasoning scenarios, where rules are often implicit, incomplete, or expressed in natural language.
>
> **A2:** We'd like to address this comment from two perspectives. First, we will extend the evaluation of our method to an autonomous driving task, which provides a more realistic testbed. Second, we will present an example of how symbolic rules can be inferred from natural-language (NL) descriptions. Taken together, these two additions demonstrate the potential of our method for real-world scenarios.
>
> **Perspective 1: Evaluation on an autonomous driving task.**
>
> We perform experiments with the setup below:
>
> - **Task:** We adopt a multi-label autonomous driving task for studying visual reasoning in high-stakes scenarios. The goal is to infer what actions out of *forward, stop, left, right* are safe depending on what objects (*e.g.*, cars, traffic signs) are present in an input dashcam image.
> - **Dataset:** We use the SDD-OIA [1] dataset, in which each image is paired with 4 binary action labels as well as 21 binary object labels. It contains 6,820 training images, 1,464 validation images, 1,464 in-distribution test images, and 1,000 OOD images. The OOD images are generated by holding out a different subset of feasible action configurations during the split construction procedure.
> - **Rules:** The rules describe the common traffic rules used in the real world. Specifically, based on a set of binary concepts indicating the presence of different obstacles on the road, these rules specify conditions for being able to *forward* (green_light $\vee$ follow $\vee$ clear $\Rightarrow$ forward), *stop* (red_light $\vee$ stop_sign $\vee$ obstacle $\Rightarrow$ stop), and for turning *left* and *right*, as well as relationships between actions (like stop $\Rightarrow$ $\neg$forward).
> - **Evaluation metric:** Task accuracy is defined such that a sample is counted as correct only when all 4 actions are predicted correctly. Concept accuracy is defined such that a sample is counted as correct only when all 21 objects are predicted correctly.
>
> The performance of our method and baselines is reported in **Table 1**.

---

> ### Author Response · Authors · 2026-04-22
> **Response to Q2 (part2)**
>
> **Table 1:** Task accuracy of different reasoning paradigms on the SDD-OIA task under covariate shifts. All paradigms adopt a 7B VLM. Results are averaged over 5 random seeds (mean $\pm$ standard deviation). The best results are highlighted in bold, and the second-best results are underlined.
>
> | Paradigm         | Task Acc on Test Set (%) | Task Acc on OOD Set (%) |
> | ---------------- | ------------------------ | ----------------------- |
> | End2end RS       | 15.37 $\pm$ 0.12         | 0.98 $\pm$ 0.15         |
> | End2end FT       | $\underline{67.92 \pm 0.08}$  | 15.00 $\pm$ 0.20        |
> | Prism            | 15.64 $\pm$ 0.27         | 17.08 $\pm$ 0.40        |
> | ViperGPT         | 14.09 $\pm$ 0.17         | $\underline{36.17 \pm 0.50}$ |
> | VLC              | 14.44 $\pm$ 0.12         | 5.20 $\pm$ 0.10         |
> | VLC (fine-tuned) | **69.38 $\pm$ 0.26**     | **43.22 $\pm$ 0.16**    |
>
> The results show that the end-to-end fine-tuning paradigm suffers from a substantial generalization gap between the test and OOD sets, suggesting that it may learn incorrect or incomplete rules. By contrast, the compositional paradigms are relatively more stable under covariate shift. For instance, Prism obtains 15.64% task accuracy on the test set and 17.08% on the OOD set.
>
> The results are also consistent with our ablation findings—fine-tuning effectively improves concept recognition performance on both the test and OOD sets. Specifically, after fine-tuning on the concept labels in the training data, the concept accuracy of the VLM increases from 0% to 37.33% on the test set and from 0% to 31.92% on the OOD set. Benefiting from this, our method with a fine-tuned VLM achieves the highest task accuracy.
>
> Overall, **these experimental results on a task with real-world rules suggest that the end-to-end fine-tuning paradigm struggles to learn the underlying prediction rules from data and therefore may not effectively improve OOD reasoning performance; nevertheless, fine-tuning remains useful for OOD concept recognition, and neuro-symbolic paradigms would benefit from this.**
>
> **Perspective 2: Symbolic rules inference from NL descriptions.**
>
> While this is still a challenging open problem, here we’d like to take an initial step to explore how simple prompting-based approaches may help and then discuss other promising directions. The experimental setup for these approaches is listed below.
>
> - **Strategies.** We consider four prompting strategies: **(i)** zero-shot prompting, where we directly prompt an LLM to generate a Boolean formula from the NL description; **(ii)** few-shot prompting, where we additionally provide one example; **(iii)** zero-shot chain-of-thought (CoT) prompting, where we ask the LLM to reason through useful intermediate Boolean conditions and introduce intermediate variables when helpful; and **(iv)** few-shot CoT prompting.
> - **Method.** For each prompting strategy, we use not only the canonical NL description from our paper but also four paraphrased versions, resulting in five NL descriptions in total. We then prompt the LLM (specifically, Qwen2.5-7B-Instruct) separately with each of the five descriptions.
> - **Evaluation metrics.** We use two metrics: **(i)** validity percentage, the percentage of generated outputs that can be successfully compiled into an SDD; and **(ii)** truth table alignment, for valid outputs, the percentage of input assignments for which the generated rule matches the ground-truth symbolic rule.

---

> ### Author Response · Authors · 2026-04-22
> **Response to Q2 (part3)**
>
> **Table 2:** Performance of different prompting strategies for symbolic rule inference from NL descriptions.
>
> |                | Zero-Shot Prompting |                       | Few-Shot Prompting  |                       | Zero-Shot CoT Prompting |                       | Few-Shot CoT Prompting |                       | Average             |                       |
> | -------------- | ------------------- | --------------------- | ------------------- | --------------------- | ----------------------- | --------------------- | ---------------------- | --------------------- | ------------------- | --------------------- |
> | Dataset        | Validity Percentage | Truth Table Alignment | Validity Percentage | Truth Table Alignment | Validity Percentage     | Truth Table Alignment | Validity Percentage    | Truth Table Alignment | Validity Percentage | Truth Table Alignment |
> | MNAdd-3dgt     | 40.00%              | 0.49%                 | 80.00%              | 0.01%                 | 80.00%                  | 5.69%                 | 60.00%                 | 0.00%                 | 65.00%              | 1.55%                 |
> | MNLogic-3dgt   | 100.00%             | 80.00%                | 80.00%              | 75.00%                | 100.00%                 | 47.50%                | 100.00%                | 85.00%                | 95.00%              | 71.88%                |
> | KandLogic-3obj | 80.00%              | 45.31%                | 100.00%             | 59.38%                | 100.00%                 | 44.38%                | 100.00%                | 59.38%                | 95.00%              | 52.11%                |
> | SDD-OIA        | 100.00%             | 50.70%                | 100.00%             | 39.48%                | 80.00%                  | 19.86%                | 100.00%                | 37.33%                | 95.00%              | 36.84%                |
>
> **Table 2** summarizes the results. From the zero-shot prompting results, we observe three main trends. **(a)** For relatively simple tasks such as MNLogic-3dgt, the LLM can infer symbolic rules with high quality, achieving 80% truth table alignment. **(b)** For more complex tasks, such as KandLogic-3obj and SDD-OIA, the validity percentage or truth table alignment drops substantially. **(c)** For tasks such as MNAdd-3dgt, which require the model to utilize its internal knowledge (e.g., full adder logic), performance becomes extremely poor.
>
> The other prompting strategies help to different extents, depending on the task. For example, few-shot prompting improves performance on KandLogic-3obj, zero-shot CoT helps on MNAdd-3dgt, and few-shot CoT gives the best results on MNLogic-3dgt.
>
> More advanced approaches may improve this further. For example, recent work [2] suggests that targeted training datasets and fine-tuning strategies (e.g., RLHF) can effectively improve a model's ability to infer symbolic structure from NL descriptions.
>
> Overall, these results suggest that **appropriate prompting often enables the LLM to produce a valid symbolic rule, but generating the correct rule remains much more difficult.** We therefore view automatic rule acquisition as an important and largely orthogonal research direction that requires further study. We will expand the above discussion in the revised paper, making it clearer in the limitations and future work sections.
>
> [1] Bortolotti, Samuele, et al. "A neuro-symbolic benchmark suite for concept quality and reasoning shortcuts." *Advances in neural information processing systems* 37 (2024): 115861-115905.
>
> [2] Yang, Yuan, et al. "Harnessing the power of large language models for natural language to first-order logic translation." *Proceedings of the 62nd Annual Meeting of the Association for Computational Linguistics (Volume 1: Long Papers)*. 2024.

---

> ### Author Response · Authors · 2026-04-22
> **Response to Q3**
>
> > **Q3:** Strengthen the empirical analysis of generalization beyond object-count shifts. The current evaluation focuses on increasing the number of objects per image while keeping the reasoning function fixed. This is a meaningful form of covariate shift, but it represents only one axis of distribution shift. To strengthen the robustness claims, the authors should include at least one additional shift type (e.g., visual appearance, object style, noise, occlusion, or concept ambiguity).
>
> **A3:** Following the helpful suggestion, we consider additional types of distribution shifts beyond object-count changes and include the corresponding experimental results. The new experimental settings for different tasks are described as follows:
>
> - MNAdd task. Starting from the MNAdd-3dgt test set, we construct two additional OOD variants: **(i) Color:** all digits are changed to red; **(ii) Rotation:** each digit image is rotated by 10 degrees clockwise.
> - MNLogic task. Starting from the MNLogic-3dgt test set, we construct two additional OOD variants: **(i) Color:** all digits are changed to red; **(ii) Rotation:** each digit image is rotated by 10 degrees clockwise.
> - KandLogic task. Starting from the KandLogic-3obj test set,  we construct two additional OOD variants: **(i) Color:** each object is assigned a color that is unseen during training; **(ii) Shape:** each object is assigned a shape that is unseen during training.
>
> **Table 3** reports the task accuracy of different reasoning paradigms under these additional distribution shifts.
>
> **Table 3:** Task accuracy (%) of different reasoning paradigms on the MNAdd, MNLogic, and KandLogic tasks under new types of covariate shifts. VLC (fine-tuned) refers to our method with a VLM fine-tuned for better concept recognition. All paradigms adopt a 7B VLM. Results are averaged over 5 random seeds (mean $\pm$ standard deviation).
>
> | Paradigm         |       MNAdd test |      MNAdd color |   MNAdd rotation |     MNLogic test |    MNLogic color | MNLogic rotation |   KandLogic test |  KandLogic color |  KandLogic shape |
> | ---------------- | ---------------: | ---------------: | ---------------: | ---------------: | ---------------: | ---------------: | ---------------: | ---------------: | ---------------: |
> | End2end RS       | 26.99 $\pm$ 0.13 | 24.57 $\pm$ 0.14 | 13.77 $\pm$ 0.07 | 71.73 $\pm$ 0.16 | 68.65 $\pm$ 0.26 | 67.79 $\pm$ 0.22 | 65.05 $\pm$ 0.09 | 61.83 $\pm$ 0.16 | 61.85 $\pm$ 0.18 |
> | End2end FT       | 79.65 $\pm$ 0.04 | 78.77 $\pm$ 0.07 | 63.57 $\pm$ 0.09 | 99.83 $\pm$ 0.00 | 99.43 $\pm$ 0.00 | 99.70 $\pm$ 0.00 | 99.97 $\pm$ 0.00 | 79.83 $\pm$ 0.02 | 72.88 $\pm$ 0.04 |
> | Prism            | 50.22 $\pm$ 0.13 | 51.33 $\pm$ 0.18 | 35.47 $\pm$ 0.23 | 51.06 $\pm$ 0.18 | 52.03 $\pm$ 0.22 | 51.87 $\pm$ 0.17 | 58.30 $\pm$ 0.10 | 57.40 $\pm$ 0.18 | 59.05 $\pm$ 0.30 |
> | ViperGPT         | 19.71 $\pm$ 0.04 |  0.60 $\pm$ 0.00 |  0.60 $\pm$ 0.00 | 85.66 $\pm$ 0.11 | 73.12 $\pm$ 0.02 | 73.12 $\pm$ 0.02 | 96.56 $\pm$ 0.08 | 50.18 $\pm$ 0.02 | 50.20 $\pm$ 0.00 |
> | VLC              | 54.62 $\pm$ 0.06 | 56.21 $\pm$ 0.14 | 39.35 $\pm$ 0.04 | 97.37 $\pm$ 0.04 | 96.17 $\pm$ 0.06 | 97.79 $\pm$ 0.04 | 95.97 $\pm$ 0.07 | 91.51 $\pm$ 0.04 | 77.13 $\pm$ 0.11 |
> | VLC (fine-tuned) | 54.62 $\pm$ 0.06 | 84.85 $\pm$ 0.05 | 72.45 $\pm$ 0.09 | 97.37 $\pm$ 0.04 | 99.67 $\pm$ 0.00 | 99.68 $\pm$ 0.02 | 95.97 $\pm$ 0.07 | 98.11 $\pm$ 0.04 | 80.62 $\pm$ 0.10 |
>
> For MNAdd and MNLogic tasks, the end-to-end fine-tuning paradigm achieves OOD performance that is close to its in-distribution test performance, suggesting that the statistical features learned by the fine-tuned VLM for these reasoning tasks are largely preserved under the studied OOD variations.
>
> In contrast, for the KandLogic task, the end-to-end fine-tuning paradigm shows a clear drop from the in-distribution test sets to the OOD sets. This indicates that the statistical features it relies on for reasoning do not transfer as well under these shifts. By comparison, both before and after fine-tuning the VLM for concept recognition, our method exhibits a smaller performance gap and maintains high task accuracy on this dataset.

---

> ### Author Response · Authors · 2026-04-22
> **Response to Q4**
>
> > **Q4:** Add an explicit analysis of error propagation from perception to reasoning. Since the proposed VLC pipeline relies on VLM-based concept recognition followed by deterministic symbolic reasoning, overall performance is tightly coupled to recognition accuracy. A short analysis quantifying how reasoning accuracy degrades under controlled recognition errors (e.g., simulated noise in concept predictions) would improve understanding of system robustness and practical deployment behavior.
>
> **A4:** Thanks for noticing this characteristic of VLC. Yes, due to the compositional nature of architecture and that its symbolic module encodes the true reasoning function, the overall performance of VLC is naturally tied to the concept recognition quality of its neural module. Section 5.2 has provided partial empirical evidence.
>
> To further support the claim, here we include an additional controlled error-propagation analysis. Starting from the ground-truth concept labels, we inject synthetic noise into the concept vector at different corruption levels and then feed the corrupted concepts into the same compiled circuit, measuring the resulting degradation in task accuracy.
>
> - **Experiment protocol.** For each test sample, we start from the ground-truth concept vector $c$. For each noise level $p$, we repeat the following procedure 5 times: independently flip each concept with probability $p$, obtain a corrupted concept vector $\tilde{c}$, feed $\tilde{c}$ into the compiled circuit to obtain the prediction $\hat{y}$, and then compute both concept accuracy and task accuracy.
>
> **Table 4** reports both concept accuracy and downstream task accuracy under varying corruption rates.
>
> **Table 4:** Concept accuracy and task accuracy of VLC under different levels of synthetic concept corruption.
>
> | Dataset        |    p |   Concept Acc (%) |      Task Acc (%) |
> | -------------- | ---: | ----------------: | ----------------: |
> | MNAdd-3dgt     | 0.00 | 100.00 $\pm$ 0.00 | 100.00 $\pm$ 0.00 |
> |                | 0.01 |  82.27 $\pm$ 0.44 |  82.33 $\pm$ 0.45 |
> |                | 0.05 |  35.88 $\pm$ 0.80 |  36.40 $\pm$ 0.73 |
> |                | 0.10 |  12.39 $\pm$ 0.65 |  13.25 $\pm$ 0.66 |
> |                | 0.20 |   1.02 $\pm$ 0.07 |   1.47 $\pm$ 0.19 |
> |                | 0.30 |   0.05 $\pm$ 0.04 |   0.24 $\pm$ 0.10 |
> | MNLogic-3dgt   | 0.00 | 100.00 $\pm$ 0.00 | 100.00 $\pm$ 0.00 |
> |                | 0.01 |  96.99 $\pm$ 0.29 |  97.02 $\pm$ 0.27 |
> |                | 0.05 |  85.57 $\pm$ 0.28 |  86.35 $\pm$ 0.27 |
> |                | 0.10 |  72.59 $\pm$ 0.41 |  75.29 $\pm$ 0.48 |
> |                | 0.20 |  50.04 $\pm$ 1.06 |  59.85 $\pm$ 1.16 |
> |                | 0.30 |  33.47 $\pm$ 0.76 |  52.37 $\pm$ 1.49 |
> | KandLogic-3obj | 0.00 | 100.00 $\pm$ 0.00 | 100.00 $\pm$ 0.00 |
> |                | 0.01 |  90.50 $\pm$ 0.38 |  95.98 $\pm$ 0.41 |
> |                | 0.05 |  61.68 $\pm$ 0.52 |  84.04 $\pm$ 0.39 |
> |                | 0.10 |  36.43 $\pm$ 0.53 |  72.28 $\pm$ 0.63 |
> |                | 0.20 |  13.12 $\pm$ 0.41 |  58.61 $\pm$ 0.72 |
> |                | 0.30 |   4.26 $\pm$ 0.56 |  52.53 $\pm$ 0.67 |
> | SDD-OIA        | 0.00 | 100.00 $\pm$ 0.00 | 100.00 $\pm$ 0.00 |
> |                | 0.01 |  81.45 $\pm$ 0.71 |  93.47 $\pm$ 0.64 |
> |                | 0.05 |  36.31 $\pm$ 1.48 |  71.90 $\pm$ 1.47 |
> |                | 0.10 |  12.36 $\pm$ 1.12 |  53.48 $\pm$ 1.03 |
> |                | 0.20 |   1.26 $\pm$ 0.35 |  32.16 $\pm$ 0.60 |
> |                | 0.30 |   0.07 $\pm$ 0.05 |  21.35 $\pm$ 0.85 |
>
> When no noise is injected, both concept accuracy and task accuracy are 100%. This is expected, because VLC’s symbolic module compiles the true reasoning function. As the corruption level increases, concept accuracy decreases, and task accuracy decreases accordingly. This validates that the performance of VLC on downstream tasks depends on the concept recognition quality of its neural module.
>
> Overall, this controlled analysis gives a clearer picture of error propagation in VLC: **the symbolic module itself does not introduce additional approximation error, and the degradation in final task performance is driven by errors in concept recognition.** We believe this analysis improves the understanding of VLC’s robustness and practical deployment behavior.
>
> Once again, thanks for your thoughtful feedback, which enables us to appropriately scope our claims and gain deeper insights into the internal error-propagation mechanism and practical feasibility of our proposed method!

---

> > ### Comment · Reviewer_vW7D · 2026-05-12
> >
> > The authors have addressed all of my concerns during the rebuttal process. I hope the clarifications and additional details provided in the rebuttal will be incorporated into the final version of the paper.

---

> > > ### Author Response · Authors · 2026-05-17
> > >
> > > Thank you for your time and effort in reviewing our responses! We are glad that our responses have addressed your concerns. We will incorporate the clarifications and additional details from the rebuttal into the final version of the paper.

---

### Review · Reviewer_gLh7 · 2026-04-09

**Summary Of Contributions:**

This work proposes a two-phase neurosymbolic pipeline for logical reasoning over images, including a VLM for concept recognition and a complied Boolean circuit that executes the symbolic rules. The method is evaluated on 3 synthetic visual reasoning tasks, spanning arthmetic addition, logical XOR< and a realtional check over geometric primitives.

## Strengths:
1. The research problem itself is well-motivated: whether VLMs can learn and apply explicit reasoning rules robustly under distribution shifts.
2. The synthetic tasks, despite simple, allow for an isolated yet challenging evaluation over the specified logical reasoning scenarios.
3. Overall, the paper is well-written with good structure.
4. The inclusion of both LLM-based and neurosymbolic methods stengthens the comparison.

## Weaknesses:
1. The two phases in the pipeline is independent and conceptually not novel. The circuit does not know or care about the inputs from the VLM. There is no connection between the two stages. If the VLM output is uncertain or even not accurate, the circuit cannot use the logical constraints to disambiguite or aid the recognition. Similar two-stage pipelines, as cited in the paper, are common in many neurosymbolic methods that adopt neural modules for perception and a symbolic solver for reasoning. The specific combination of prompting a VLM for concept extraction and then running a compiled circuit is straightforward.
2. The proposed method assumes that symbolic rules are provided in an exact, compilable form. This is a very strong assumption that rarely holds in practice. Accordingly, the tasks evaluated are synthetic with weak connections to real-world applications, like VQA, in which a clean formalization of the reasoning function is difficult.
3. The introduction motivates the work broadly around robust VLM reasoning, but the proposed method operates under narrow assumptions. The positioning should be revised to better reflect the restricted scope.
4. The comparison with ViperGPT and Prism appears unfair. For example, the visual detection model in ViperGPT, GroundingDINO, is not designed for the specific visual formats in these tasks like digit recognition. I wonder if the performance gap can be significantly reduced if their visual detectors are also fine-tuned with 3dgt/obj datasets, or if the same VLM, Qen2.5-VL-7B, prompted with the same task-specific few-shot examples as VLC, is used as the detection model for ViperGPT.
5. As also stated in the paper, " the performance of VLC on reasoning tasks is dependent solely on the VLM’s recognition capability." The current improvements mainly stem from the task decomposition and the better fine-tuning of the VLM for the first stage recognition, rather than a good design of the neurosymbolic pipeline.

**Audience:**

Yes

**Audience Explanation:**

The paper indeed addresses a timely and important research problem, which can be of interest to the broader community.

**Claims And Evidence:**

No

**Claims Explanation:**

See Weaknesses above.

**Requested Changes:**

1. Provide a more balanced discussion of the ViperGPT comparison with better visual recognition model. The current poor performance appears driven by a detection model mismatch rather than a paradigm-level failure.
2. Report wall-clock time and computational cost for each paradigm for a better understanding of the trade-offs.
3. Discuss or add an experiment with how the approach scales when the symbolic rules become more complex, e.g., when multiple logical rules need to compose.
4. If possible, evaluate with one additional VLM family beyond Qwen2.5-VL to demonstrate the cross-model generalizability.
5. Revise or soften the claims about VLC achieving "strong performance" and "robust reasoning" in the abstract and introduction with more accurate positioning.

---

> ### Author Response · Authors · 2026-04-22
> **Response to Q1**
>
> We sincerely thank the reviewer for the constructive comments! We're glad you found the research problem well-motivated and the paper well-written with comprehensive experiments!
>
> > **Q1:** Discuss or add an experiment with how the approach scales when the symbolic rules become more complex, e.g., when multiple logical rules need to compose.
>
> **A1:** First, we'd like to clarify that, in our paper, we focus on relatively simple rules given the following concerns:
>
> - Generalization can be controlled in a rigorous way.
> - We want to show that even under these simple rules, end-to-end fine-tuned VLMs fail to generalize to OOD samples, thus suggesting that end-to-end fine-tuning does not reliably learn the underlying reasoning rules from data.
>
> However, we agree with the reviewer that including experiments with more complex rules can better support our claims. To this end, we consider an additional task with real-world rules. The experimental settings are listed below:
>
> - **Task:** We adopt a multi-label autonomous driving task for studying visual reasoning in high-stakes scenarios. The goal is to infer what actions out of *forward, stop, left, right* are safe depending on what objects (*e.g.*, cars, traffic signs) are present in an input dashcam image.
> - **Dataset:** We use the SDD-OIA [1] dataset, in which each image is paired with 4 binary action labels as well as 21 binary object labels. It contains 6,820 training images, 1,464 validation images, 1,464 in-distribution test images, and 1,000 OOD images. The OOD images are generated by holding out a different subset of feasible action configurations during the split construction procedure.
> - **Rules:** The rules describe the common traffic rules used in the real world. Specifically, based on a set of binary concepts indicating the presence of different obstacles on the road, these rules specify conditions for being able to *forward* (green_light $\vee$ follow $\vee$ clear $\Rightarrow$ forward), *stop* (red_light $\vee$ stop_sign $\vee$ obstacle $\Rightarrow$ stop), and for turning *left* and *right*, as well as relationships between actions (like stop $\Rightarrow$ $\neg$forward).
> - **Evaluation metric:** Task accuracy is defined such that a sample is counted as correct only when all 4 actions are predicted correctly. Concept accuracy is defined such that a sample is counted as correct only when all 21 objects are predicted correctly.
>
> The performance of our method and baselines is reported in **Table 1**.
>
> **Table 1:** Task accuracy of different reasoning paradigms on the SDD-OIA task under covariate shifts. VLC (fine-tuned) refers to our method with a VLM fine-tuned for better concept recognition. All paradigms adopt a 7B VLM. Results are averaged over 5 random seeds (mean $\pm$ standard deviation). The best results are highlighted in bold, and the second-best results are underlined.
>
> | Paradigm         | Task Acc on Test Set (%) | Task Acc on OOD Set (%) |
> | ---------------- | ------------------------ | ----------------------- |
> | End2end RS       | 15.37 $\pm$ 0.12         | 0.98 $\pm$ 0.15         |
> | End2end FT       | $\underline{67.92 \pm 0.08}$  | 15.00 $\pm$ 0.20        |
> | Prism            | 15.64 $\pm$ 0.27         | 17.08 $\pm$ 0.40        |
> | ViperGPT         | 14.09 $\pm$ 0.17         | $\underline{36.17 \pm 0.50}$ |
> | VLC              | 14.44 $\pm$ 0.12         | 5.20 $\pm$ 0.10         |
> | VLC (fine-tuned) | **69.38 $\pm$ 0.26**     | **43.22 $\pm$ 0.16**    |
>
> The results show that the end-to-end fine-tuning paradigm suffers from a substantial generalization gap between the test and OOD sets, suggesting that it may learn incorrect or incomplete rules. By contrast, the compositional paradigms are relatively more stable under covariate shift. For instance, Prism obtains 15.64% task accuracy on the test set and 17.08% on the OOD set.
>
> The results are also consistent with our ablation findings—fine-tuning effectively improves concept recognition performance on both the test and OOD sets. Specifically, after fine-tuning on the concept labels in the training data, the concept accuracy of the VLM increases from 0% to 37.33% on the test set and from 0% to 31.92% on the OOD set. Benefiting from this, our method with a fine-tuned VLM achieves the highest task accuracy.
>
> Overall, **these experimental results on a task with real-world rules suggest that the end-to-end fine-tuning paradigm struggles to learn the underlying prediction rules from data and therefore may not effectively improve OOD reasoning performance; nevertheless, fine-tuning remains useful for OOD concept recognition, and neuro-symbolic paradigms would benefit from this.**
>
> [1] Bortolotti, Samuele, et al. "A neuro-symbolic benchmark suite for concept quality and reasoning shortcuts." *Advances in neural information processing systems* 37 (2024): 115861-115905.

---

> ### Author Response · Authors · 2026-04-22
> **Response to Q2**
>
> > **Q2:** If possible, evaluate with one additional VLM family beyond Qwen2.5-VL to demonstrate the cross-model generalizability.
>
> **A2:** To demonstrate the cross-model generalizability of the proposed method, we further evaluate both our method and the baselines using an additional VLM family. Specifically, we use **InternVL3_5-8B-Instruct** as the backbone VLM. The results are reported in **Table 5**. P.S. Due to the substantially higher inference time of ViperGPT, we have not yet obtained a complete set of InternVL results within the rebuttal period, and therefore do not report it here.
>
> **Table 5:** Task accuracy of different reasoning paradigms on the MNAdd, MNLogic, and KandLogic tasks under covariate shifts. All paradigms adopt a 7B VLM, InternVL3_5-8B-Instruct specifically. The best results are highlighted in bold, and the second-best results are underlined.
>
> | Task       |        MNAdd |        MNAdd |        MNAdd |      MNLogic |      MNLogic |      MNLogic |    KandLogic |    KandLogic |    KandLogic |
> | ---------- | -----------: | -----------: | -----------: | -----------: | -----------: | -----------: | -----------: | -----------: | -----------: |
> | Paradigm   |         3dgt |         5dgt |         7dgt |         3dgt |         5dgt |         7dgt |         3obj |         5obj |         7obj |
> | End2end RS |         0.90 |         0.10 |         0.13 |        43.23 | $\underline{53.13}$ |        50.53 |        59.50 |        78.70 |        51.90 |
> | End2end FT |    **92.67** |        37.93 |        13.47 |    **99.90** |        47.87 | $\underline{53.57}$ |    **99.97** | $\underline{95.30}$ | $\underline{97.67}$ |
> | Prism      |        36.43 | $\underline{43.73}$ | $\underline{39.90}$ |        50.00 |        51.37 |        51.13 |        68.70 |        79.97 |        57.37 |
> | VLC        | $\underline{85.90}$ |    **82.60** |    **78.70** | $\underline{95.87}$ |    **95.37** |    **87.43** | $\underline{99.27} $|    **97.77** |    **97.97** |
>
> These results broadly reproduce the main findings of Section 5.2: VLC remains the most consistent paradigm under covariate shifts across tasks. In particular, VLC achieves high OOD task accuracy on MNAdd (82.60/78.70 on 5dgt/7dgt), MNLogic (95.37 on 5dgt and 87.43 on 7dgt), and KandLogic (97.77/97.97 on 5obj/7obj).
>
> At the same time, these results also provide a more nuanced view of end-to-end fine-tuning. This paradigm generalizes well on KandLogic, while showing substantial degradation on MNAdd and MNLogic. This suggests that end-to-end fine-tuning may generalize in some cases, but does not do so reliably across tasks under covariate shifts.
>
> We also observe that Prism remains clearly less consistent than VLC. Although it improves over end-to-end reasoning in some settings, such as MNAdd, it does not maintain uniformly high accuracy across tasks, since its performance still depends on the reasoning ability of a black-box LLM.
>
> Overall, the additional InternVL experiments further support our main claim that, when the reasoning rules are explicitly provided, **structurally encoding the true reasoning function within a circuit and then incorporating this circuit within the overall architecture leads to more reliable OOD generalization under covariate shifts.**

---

> ### Author Response · Authors · 2026-04-22
> **Responses to Q3 and Q4**
>
> > **Q3:** Report wall-clock time for each paradigm for a better understanding of the trade-offs.
>
> **A3:** **Table 6** reports the inference time of each reasoning paradigm on different tasks. All evaluations are conducted on a single NVIDIA A100-SXM4 GPU with 80 GB memory.
>
> **Table 6:** Inference time (seconds) of different reasoning paradigms on the MNAdd-3dgt, MNLogic-3dgt, and KandLogic-3dgt test sets.
>
> | Paradigm   | MNAdd-3dgt | MNLogic-3dgt | KandLogic-3dgt |
> | ---------- | ---------- | ------------ | -------------- |
> | End2end RS | 207.67     | 211.28       | 432.43         |
> | End2end FT | 40.34      | 38.24        | 50.02          |
> | Prism      | 250.92     | 238.59       | 522.66         |
> | ViperGPT   | 2106.14    | 1274.43      | 3605.11        |
> | VLC        | 201.52     | 198.39       | 451.64         |
>
> We observe that the end-to-end fine-tuned VLM (End2end FT) is the fastest on all three datasets. This is because, after fine-tuning, the model can directly generate outputs in the desired format and therefore does not require few-shot examples in the prompt.
>
> Compared with it, the pretrained VLM (End2end RS) and VLC have similar inference times, as they both rely on five few-shot examples in the prompt: the former uses them to demonstrate the reasoning output format, while the latter uses them to demonstrate the concept recognition output format.
>
> The longer inference time of Prism than VLC suggests that, on the studied datasets, LLM-based reasoning is slower than circuit-based reasoning. ViperGPT is by far the slowest paradigm because it may invoke various pretrained models multiple times for each sample during inference, which leads to substantially higher runtime.
>
> Overall, these results reveal a trade-off between flexibility and efficiency: **paradigms with greater flexibility tend to be less efficient**. For example, compared with Prism, ViperGPT is more flexible because it can invoke a wider range of pretrained models many times, but it uses higher runtime. Compared with VLC, Prism is more flexible because it relies on the general reasoning ability of an LLM rather than strictly following predefined symbolic rules, while it uses higher runtime.
>
> > **Q4:** Provide a more balanced discussion of the ViperGPT comparison with a better visual recognition model. The current poor performance appears driven by a detection model mismatch rather than a paradigm-level failure. I wonder if the performance gap can be significantly reduced if the same VLM, Qwen2.5-VL-7B-Instruct, prompted with the same task-specific few-shot examples as VLC, is used as the detection model for ViperGPT.
>
> **A4:** Thanks for noticing this. First, we'd like to clarify that in our implementation, ViperGPT actually uses the same visual recognition model as VLC and Prism (i.e., Qwen2.5-VL-7B-Instruct), while relying on GroundingDINO as the detection model, as stated in Appendix C.
>
> Next, we'd like to explain how ViperGPT uses these models. In the prompt that ViperGPT uses to prompt the code generation model, they define a `find(object_name)` function that returns image patches that match the queried object. This function would call the detection model. They also define a `simple_query(question)` function that returns the answer to a basic question asked about the image patch. This function would call the visual recognition model.
>
> Therefore, if we wanted ViperGPT to rely *only* on Qwen2.5-VL-7B-Instruct, we have to redefine the `find(object_name)` function so that the bounding boxes are given by a VLM, or redesign the prompt so that the generated program uses only `simple_query(question)`-style operations.
>
> Here, we'd like to interpret the reviewer's concern as: *What if we use a code program, rather than a circuit, in the proposed pipeline? How does such a code-program-based reasoning paradigm perform?* Actually, at the level of rule expressivity, both a correct code program and a circuit can represent the same deductive rule. Their downstream task performance is thus supposed to be similar. Due to the rebuttal time limit, we have not yet completed a full set of experiments along this direction. But in the revision, we will include the investigation of this point, making the performance of code-program-based reasoning paradigms more balanced.

---

> ### Author Response · Authors · 2026-04-22
> **Response to Q5**
>
> > **Q5:** Revise or soften the claims about VLC achieving "strong performance" and "robust reasoning" in the abstract and introduction with more accurate positioning.
>
> **A5:** We will refine the statements in the paper to make sure the claims are properly scoped and accurately interpreted by readers. In particular,
>
> - Abstract. We will revise "Experiments on three visual deductive reasoning tasks with distinct rule sets show that VLC consistently achieves strong performance under covariate shifts, highlighting its ability to support robust reasoning." **to** "Experiments on three *simple* visual deductive reasoning tasks with distinct rule sets show that VLC consistently achieves higher task accuracy on out-of-distribution data than other reasoning paradigms."
> - Section 1 (Introduction). We will add a footnote to clarify the reasoning robustness in our context: "Throughout this paper, we use reasoning robustness to refer specifically to the ability to apply a fixed, explicitly provided deductive rule under controlled covariate shifts in perceptual input, rather than to general-purpose reasoning in open-ended multimodal settings."
> - Section 6 (Conclusions).
>   - We will revise "In this paper, we study the problem of robust VLM reasoning and present a neuro-symbolic investigation." **to** "In this paper, we study the robustness of rule-based deductive reasoning in VLMs under controlled covariate shifts and present a neuro-symbolic investigation."
>   - We will also clarify in the limitation paragraph that our conclusions are restricted to visual deductive reasoning with explicit task rules and controlled covariate shifts, and do not directly establish robustness in open-ended multimodal reasoning settings.
>
> Overall, the main takeaway we want to deliver is: *when reasoning rules are explicitly provided, encoding them into a symbolic module, rather than learning them through end-to-end fine-tuning, can more reliably improve reasoning performance under covariate shifts.* We hope these revisions could make the scope of our claims clearer and help readers interpret our contributions more accurately.
>
> Once again, thanks for your constructive comments, which enable us to appropriately scope our claims, gain deeper insights into the trade-offs of different reasoning paradigms, and demonstrate the cross-model generalizability of our proposed method!

---

> ### Comment · Reviewer_gLh7 · 2026-05-16
>
> Thanks for the responses and the additional experiments. The Q5 revisions read well, and I retract my Q4 concern after the clarification on ViperGPT setups. A few concerns remain:
> 1. Contribution of the circuit (W1, W5). The SDD-OIA results strengthen my W5: the jump from 5.20% to 43.22% OOD comes from fine-tuning the VLM, and without concept supervision VLC underperforms Prism on OOD. Combined with W1, this suggests the paradigm's gains come primarily from (a) task decomposition and (b) concept-level supervision, rather than on the symbolic component itself. I would like to see this engaged with directly, either by acknowledging it as a scope limitation or by an ablation isolating the circuit's contribution.
> 2. Code-program vs. circuit (Q4). The authors' proposed reframing would directly address the point above. I strongly encourage including it in the revision.
> 3. Minor (Q2). End2end FT with InternVL appears to generalize better on KandLogic 5obj/7obj than Qwen, suggesting some backbone-dependence worth a brief discussion.

---

> > ### Author Response · Authors · 2026-05-17
> >
> > Thank you for carefully reviewing our responses and providing valuable follow-up comments! We address each remaining concern below.
> >
> > **Q5 revisions.** We appreciate your acknowledgment that the Q5 revisions read well. We have incorporated the corresponding revisions into the paper as discussed in R5.
> >
> > **Contribution of the circuit (W1, W5).** Thank you for the insightful observation on our experimental results. In the revision, we will explicitly attribute the observed performance gains to (a) task decomposition and (b) concept-level supervision, and acknowledge the contribution of the circuit as a scope limitation. We believe this clarification will help readers more accurately interpret the source of VLC's improvements.
> >
> > **Code-program vs. circuit (Q4).** Thank you for acknowledging our reframing of your concern. We will make sure to include the described experiments in the revision.
> >
> > **Minor (Q2).** Thank you for noticing that the End2end FT baseline shows different generalization behavior on the KandLogic task when using different VLM backbones. We will include the experiments from R2 in the revision, explicitly note this difference, and add a discussion on the backbone-dependence of end-to-end fine-tuning generalization.
> >
> > Thank you once again for your time and thoughtful engagement, which have helped us make the claims in the paper clearer, properly scoped, and more comprehensive!

---

### Review · Reviewer_ZTxg · 2026-04-13

**Summary Of Contributions:**

This paper introduces a simple yet effective neuro-symbolic framework that integrates the strengths of vision-language models in concept recognition with circuit-based symbolic reasoning. Despite being training-free, the proposed method achieves improved performance compared to existing approaches.


### Strengths ###

1. A well-motivated research question in understanding the VLM's capability in visual deductive reasoning
2. The paper is well-written and easy to follow

### Weaknesse ###

1. The scope of the paper appears overclaimed. While the paper raises the question ‘Can VLMs Reason Robustly?’, the evaluation is limited to relatively simple synthetic visual deductive reasoning tasks, such as addition, XOR, and relational checks. I would recommend revising the title and introduction to better align the claims with the scope of the experiments.

2. It would be beneficial to include robustness evaluations, such as introducing rotated or skewed digits, partial occlusions, or more realistic inputs (e.g., scene text digits) rather than standard black-and-white digits.

3. It would be highly valuable to include evaluations on widely used models such as ChatGPT and Gemini. These models may perform well on the proposed tasks; however, this is currently unclear to the audience.

4. The paper should include a more thorough discussion of its limitations, as well as its applicability to more complex scenarios and real-world settings. Such a discussion would help clarify the broader motivation and contextualize the contributions.

**Audience:**

Yes

**Audience Explanation:**

The topic is interesting and relevant to a broad segment of the TMLR audience, particularly researchers focused on VLM reasoning. However, the reviewer is not convinced that the paper will have a greater impact, given the limited scope of the evaluation.

**Broader Impact Concerns:**

No concerns for this part.

**Claims And Evidence:**

Yes

**Claims Explanation:**

The experiments are comprehensive for the evaluated datasets.

**Requested Changes:**

Please clarify the contributions and moderate the strength of the claims.

---

> ### Author Response · Authors · 2026-04-22
> **Response to Q1**
>
> We sincerely thank the reviewer for the thoughtful feedback! We're glad you found the research problem well-motivated and the paper well-written with comprehensive experiments!
>
> > **Q1:** Please clarify the contributions and moderate the strength of the claims.
>
> **A1:** We will refine the statements in the paper to make sure the claims are properly scoped and accurately interpreted by readers. In particular,
>
> - Abstract. We will revise "Experiments on three visual deductive reasoning tasks with distinct rule sets show that VLC consistently achieves strong performance under covariate shifts, highlighting its ability to support robust reasoning." **to** "Experiments on three *simple* visual deductive reasoning tasks with distinct rule sets show that VLC consistently achieves higher task accuracy on out-of-distribution data than other reasoning paradigms."
> - Section 1 (Introduction). We will add a footnote to clarify the reasoning robustness in our context: "Throughout this paper, we use reasoning robustness to refer specifically to the ability to apply a fixed, explicitly provided deductive rule under controlled covariate shifts in perceptual input, rather than to general-purpose reasoning in open-ended multimodal settings."
> - Section 6 (Conclusions).
>   - We will revise "In this paper, we study the problem of robust VLM reasoning and present a neuro-symbolic investigation." **to** "In this paper, we study the robustness of rule-based deductive reasoning in VLMs under controlled covariate shifts and present a neuro-symbolic investigation."
>   - We will also clarify in the limitation paragraph that our conclusions are restricted to visual deductive reasoning with explicit task rules and controlled covariate shifts, and do not directly establish robustness in open-ended multimodal reasoning settings.
>
> We hope these revisions could make the scope of our claims clearer and help readers interpret our contributions more accurately.

---

> ### Author Response · Authors · 2026-04-22
> **Responses to Q2 and Q3**
>
> > **Q2:** It would be beneficial to include robustness evaluations, such as introducing rotated or skewed digits, partial occlusions, or more realistic inputs (e.g., scene text digits) rather than standard black-and-white digits.
>
> **A2:** Following the helpful suggestion, we consider additional types of distribution shifts beyond object-count changes and include the corresponding experimental results. The new experimental settings for different tasks are described as follows:
>
> - MNAdd task. Starting from the MNAdd-3dgt test set, we construct two additional OOD variants: **(i) Color:** all digits are changed to red; **(ii) Rotation:** each digit image is rotated by 10 degrees clockwise.
> - MNLogic task. Starting from the MNLogic-3dgt test set, we construct two additional OOD variants: **(i) Color:** all digits are changed to red; **(ii) Rotation:** each digit image is rotated by 10 degrees clockwise.
> - KandLogic task. Starting from the KandLogic-3obj test set,  we construct two additional OOD variants: **(i) Color:** each object is assigned a color that is unseen during training; **(ii) Shape:** each object is assigned a shape that is unseen during training.
>
> **Table 3** reports the task accuracy of different reasoning paradigms under these additional distribution shifts.
>
> **Table 3:** Task accuracy (%) of different reasoning paradigms on the MNAdd, MNLogic, and KandLogic tasks under new types of covariate shifts. VLC (fine-tuned) refers to our method with a VLM fine-tuned for better concept recognition. All paradigms adopt a 7B VLM. Results are averaged over 5 random seeds (mean $\pm$ standard deviation).
>
> | Paradigm         |       MNAdd test |      MNAdd color |   MNAdd rotation |     MNLogic test |    MNLogic color | MNLogic rotation |   KandLogic test |  KandLogic color |  KandLogic shape |
> | ---------------- | ---------------: | ---------------: | ---------------: | ---------------: | ---------------: | ---------------: | ---------------: | ---------------: | ---------------: |
> | End2end RS       | 26.99 $\pm$ 0.13 | 24.57 $\pm$ 0.14 | 13.77 $\pm$ 0.07 | 71.73 $\pm$ 0.16 | 68.65 $\pm$ 0.26 | 67.79 $\pm$ 0.22 | 65.05 $\pm$ 0.09 | 61.83 $\pm$ 0.16 | 61.85 $\pm$ 0.18 |
> | End2end FT       | 79.65 $\pm$ 0.04 | 78.77 $\pm$ 0.07 | 63.57 $\pm$ 0.09 | 99.83 $\pm$ 0.00 | 99.43 $\pm$ 0.00 | 99.70 $\pm$ 0.00 | 99.97 $\pm$ 0.00 | 79.83 $\pm$ 0.02 | 72.88 $\pm$ 0.04 |
> | Prism            | 50.22 $\pm$ 0.13 | 51.33 $\pm$ 0.18 | 35.47 $\pm$ 0.23 | 51.06 $\pm$ 0.18 | 52.03 $\pm$ 0.22 | 51.87 $\pm$ 0.17 | 58.30 $\pm$ 0.10 | 57.40 $\pm$ 0.18 | 59.05 $\pm$ 0.30 |
> | ViperGPT         | 19.71 $\pm$ 0.04 |  0.60 $\pm$ 0.00 |  0.60 $\pm$ 0.00 | 85.66 $\pm$ 0.11 | 73.12 $\pm$ 0.02 | 73.12 $\pm$ 0.02 | 96.56 $\pm$ 0.08 | 50.18 $\pm$ 0.02 | 50.20 $\pm$ 0.00 |
> | VLC              | 54.62 $\pm$ 0.06 | 56.21 $\pm$ 0.14 | 39.35 $\pm$ 0.04 | 97.37 $\pm$ 0.04 | 96.17 $\pm$ 0.06 | 97.79 $\pm$ 0.04 | 95.97 $\pm$ 0.07 | 91.51 $\pm$ 0.04 | 77.13 $\pm$ 0.11 |
> | VLC (fine-tuned) | 54.62 $\pm$ 0.06 | 84.85 $\pm$ 0.05 | 72.45 $\pm$ 0.09 | 97.37 $\pm$ 0.04 | 99.67 $\pm$ 0.00 | 99.68 $\pm$ 0.02 | 95.97 $\pm$ 0.07 | 98.11 $\pm$ 0.04 | 80.62 $\pm$ 0.10 |
>
> For MNAdd and MNLogic tasks, the end-to-end fine-tuning paradigm achieves OOD performance that is close to its in-distribution test performance, suggesting that the statistical features learned by the fine-tuned VLM for these reasoning tasks are largely preserved under the studied OOD variations.
>
> In contrast, for the KandLogic task, the end-to-end fine-tuning paradigm shows a clear drop from the in-distribution test sets to the OOD sets. This indicates that the statistical features it relies on for reasoning do not transfer as well under these shifts. By comparison, both before and after fine-tuning the VLM for concept recognition, our method exhibits a smaller performance gap and maintains high task accuracy on this dataset.
>
> > **Q3:** It would be highly valuable to include evaluations on widely used models such as ChatGPT and Gemini. These models may perform well on the proposed tasks; however, this is currently unclear to the audience.
>
> **A3:** Thank you for this helpful suggestion. We agree that evaluating widely used proprietary models such as ChatGPT and Gemini would be valuable. However, due to the limited time and resources during the rebuttal period, we were not able to conduct a careful and fair comparison on these API-based closed-source models. But, as a step toward broader model coverage, we have added results on an additional VLM family, **InternVL3_5-8B-Instruct** (please see **A2** for Reviewer gLh7). From those results, we found that our main conclusions remain consistent.
>
> Once again, thanks for your thoughtful feedback, which enables us to appropriately scope our claims and explore broader types of robustness and models!

---

### Decision · Action_Editor_uaGP · 2026-05-26

**Recommendation:** Accept with minor revision

**Additional Comments:**

I recommend acceptance with minor revision.

The final version should make the following changes before acceptance:

-   The paper should state that it studies rule-based visual deductive reasoning under controlled covariate shifts, with explicit rules available at inference time. It should avoid language that suggests general robust multimodal reasoning.
- Incorporate the rebuttal experiments and analyses into the paper, including the added covariate shifts, InternVL results, SDD-OIA results, error-propagation analysis, runtime comparison, and natural-language-to-symbolic-rule probing.
- Much of the performance gain comes from task decomposition and concept-level supervision, while the circuit mainly provides exact execution once the correct concepts and rules are available. This should be framed as a limitation of the current evidence, not as a failure of the method.
- Add the promised discussion or experiment comparing code-program-based rule execution with circuit-based rule execution. This is important because the current evidence does not fully isolate the circuit representation from more general explicit program execution.
Discuss the practical limits of the approach: exact symbolic rules are assumed, rules are task-specific, and final accuracy remains tied to concept recognition quality.

With these revisions, the paper meets the TMLR bar. The work is technically sound, clear, and useful to part of the TMLR audience, although its novelty and scope are modest.

**Audience:**

Yes

**Audience Explanation:**

At least some TMLR readers will be interested in the findings. The paper studies a timely question: whether VLMs learn reusable reasoning rules or mainly fit shortcuts under distribution shift. The controlled setup is simple, but it isolates a useful failure mode and gives clear evidence that task decomposition can improve reliability in explicit-rule settings. The results are relevant to readers working on VLM reasoning, robustness, concept bottlenecks, and neuro-symbolic methods.

**Claims And Evidence:**

Yes

**Claims Explanation:**

The main empirical claim is supported within the paper's intended setting: visual deductive reasoning with explicitly provided rules under controlled covariate shifts. The original manuscript states this claim too broadly in places, especially around "robust VLM reasoning," but the experiments do show that end-to-end fine-tuning can fit in-distribution data while failing to apply the same rule under object-count shifts, and that the proposed decomposition into concept recognition and symbolic rule execution gives more stable OOD performance.

The reviewers raised valid concerns about scope, the reliance on synthetic tasks, the assumption of exact symbolic rules, and the limited independent contribution of the circuit module. The authors addressed these concerns through additional experiments and concrete revision commitments: added shift types, an additional VLM family, SDD-OIA experiments, error-propagation analysis, runtime analysis, and discussion of natural-language-to-rule conversion. These additions make the evidence sufficient for the narrower claim that explicit rule execution can improve OOD performance when the rule is known and the concepts can be recognized. The final version must clearly reflect this narrower scope.

---

> ### Author Response · Authors · 2026-06-18
>
> We thank the AE and all reviewers for the time and effort they put into evaluating our submission. We greatly appreciate your helpful suggestions, which have made the paper more sound and clear.
>
> We have revised the manuscript to ensure that the requested changes are included. Specifically, we have: 1) refined the statements in the abstract, introduction, and conclusion to make sure that the claims are properly scoped and accurately interpreted by readers; 2) incorporated the rebuttal and promised experiments and analysis in Appendices F through I; 3) added clarification on the performance gain throughout the paper; and 4) expanded the discussion of practical limits in both the conclusion and Appendix A.
>
> Thank you again for your constructive feedback. We believe these revisions make the scope, evidence, and limitations of the work clearer.